# A novel and accurate full-length HTT mouse model for Huntington's disease

**Sushila A Shenoy[1][†], Sushuang Zheng[2]*[†], Wencheng Liu[1], Yuanyi Dai[2], Yuanxiu Liu[2], Zhipeng Hou[3], Susumu Mori[3], Yi Tang[4], Jerry Cheng[5], Wenzhen Duan[6], Chenjian Li[2]***

[1]Department of Neuroscience, Weill Cornell Graduate School of Medical Sciences, New York, United States; [2]The MOE Key Laboratory of Cell Proliferation and Differentiation, School of Life Sciences, Peking University, Beijing, China; [3]The Russell H. Morgan Department of Radiology and Radiological Sciences, Johns Hopkins University School of Medicine, Baltimore, United States; [4]Innovation Center for Neurological Disorders, Department of Neurology, Xuanwu Hospital Capital Medical University, National Center for Neurological Disorders, Beijing, China; [5]Department of Computer Science, New York Institute of Technology, New York, United States; [6]Division of Neurobiology, Department of Psychiatry and Behavioral Sciences; Solomon H.Snyder Department of Neuroscience, Johns Hopkins University School of medicine, Baltimore, United States

**\*For correspondence:**
zheng_sushuang@pku.edu.cn (SZ);
li_chenjian@pku.edu.cn (CL)

[†]These authors contributed equally to this work

**Competing interest:** The authors declare that no competing interests exist.

**Abstract** Here, we report the generation and characterization of a novel Huntington's disease (HD) mouse model BAC226Q by using a bacterial artificial chromosome (BAC) system, expressing full-length human HTT with ~226 CAG-CAA repeats and containing endogenous human HTT promoter and regulatory elements. BAC226Q recapitulated a full-spectrum of age-dependent and progressive HD-like phenotypes without unwanted and erroneous phenotypes. BAC226Q mice developed normally, and gradually exhibited HD-like psychiatric and cognitive phenotypes at 2 months. From 3 to 4 months, BAC226Q mice showed robust progressive motor deficits. At 11 months, BAC226Q mice showed significant reduced life span, gradual weight loss and exhibited neuropathology including significant brain atrophy specific to striatum and cortex, striatal neuronal death, widespread huntingtin inclusions, and reactive pathology. Therefore, the novel BAC226Q mouse accurately recapitulating robust, age-dependent, progressive HD-like phenotypes will be a valuable tool for studying disease mechanisms, identifying biomarkers, and testing gene-targeting therapeutic approaches for HD.

## Editor's evaluation

This work describes creation of a novel Huntington disease (HD) mouse model that has potential value to the HD field. This model demonstrates disease features that have previously been difficult to capture and opens new avenues of examination.

## Introduction

Huntington's disease (HD) is an autosomal-dominant hereditary neurodegenerative disorder caused by a pathogenic expansion of the CAG trinucleotide repeats in exon 1 of the huntingtin (HTT) gene (A novel gene containing a trinucleotide repeat that is expanded and unstable on Huntington's disease chromosomes. The Huntington's Disease Collaborative Research Group, 1993). The clinical features of HD include motor deficit, psychiatric disturbance, cognitive impairment, and peripheral signs such

as weight loss and sleep disturbance (*Ghosh and Tabrizi, 2015*). Disease onset, which is dependent on the CAG repeat length, is usually defined by the onset of motor symptoms, although non-motor symptoms are often present many years in advance. The striking neuropathological characteristic of HD, induced by the mutant Htt protein (mHtt), is the progressive brain atrophy mainly in the striatum and cerebral cortex (*Walker, 2007*). Despite ubiquitously expressed mHtt protein in HD brain, the primary vulnerable neuronal types are medium spiny neurons (MSNs) in the striatum and pyramidal neurons in the cortex (*Vonsattel and DiFiglia, 1998*). Widespread mHtt protein aggregation in HD brain tissue is another hallmark of HD pathology (*Davies and Scherzinger, 1997*). While the causative gene for HD was identified 28 years ago (*Macdonald, 1993*), there are still no effective disease-modifying therapies available to prevent or delay the progression of this disorder.

Since the identification of causative gene for HD, a large number of animal models have been generated in a variety of animal species including *Caenorhabditis elegans*, *Drosophila melanogaster*, zebrafish, mouse, rat, sheep, pig, and non-human primates to better elucidate the complex pathogenic mechanisms in HD, and to develop potential therapeutic strategies for HD (*Farshim and Bates, 2018*). Although large animal models, such as pig and non-human primates, are very useful to study HD (*Niu et al., 2010*; *Sasaki et al., 2009*; *Yan et al., 2018*; *Yang et al., 2008*), mouse models still dominate the research field and provide us with valuable tools to investigate the pathogenesis of HD and therapeutic targets.

To date, there are more than 20 different HD mouse models available (*Brooks and Dunnett, 2015*; *Crook and Housman, 2011*; *Farshim and Bates, 2018*; *Pouladi et al., 2013*). In general, those genetic mouse models can be classified into three groups based on distinct strategies: the transgenic models carrying human HTT N-terminal fragments with CAG expansions such as R6/1, R6/2, and N171-82Q (*Carter et al., 1999*; *Mangiarini et al., 1996*; *Schilling et al., 1999*), the transgenic full-length mutant HTT models such as YAC128, BACHD, BAC-225Q (*Gray et al., 2008*; *Slow et al., 2003*; *Wegrzynowicz et al., 2015*), humanized HD mice (Hu97/18 and Hu128/21) (*Southwell et al., 2017*; *Southwell et al., 2013*), and knock-in models with expanded CAG repeats inserted into a mouse Htt exon 1 such as HdhQ72 and HdhQ150 or into a humanized exon 1 sequence such as HdhQ111, HdhQ140, N160Q, zQ175, and Q175FDN (*Heng et al., 2007*; *Hickey et al., 2008*; *Levine et al., 1999*; *Lin et al., 2001*; *Liu et al., 2016*; *Menalled et al., 2012*; *Menalled et al., 2003*; *Shelbourne et al., 1999*; *Southwell et al., 2016*; *Wheeler et al., 1999*). The existing mouse models mimic some aspects of HD including behavioral disturbances and neuropathological changes of the disease. However, none of them can fully recapitulate human disease, and many of them have spurious phenotypes that are irrelevant or opposite to human conditions. The N-terminal fragment transgenic mice such as R6/2 show a robust phenotype and severely reduced life span, but they demonstrated severe developmental deficits and nonspecific neurodegeneration throughout the central nervous system (*Carter et al., 1999*; *Crook and Housman, 2011*; *Heng et al., 2007*; *Mangiarini et al., 1996*; *Southwell et al., 2016*) as well as diabetes, which are not correlated with HD patients. In comparison, full-length transgenic HD mice and Hdh knock-in mice replicate relatively better the neuropathology of HD; however, they display much milder phenotypes and slower progression (*Brooks et al., 2012*; *Farshim and Bates, 2018*; *Pouladi et al., 2013*). Moreover, YAC128, BACHD, and humanized HD strains (Hu97/18 and Hu128/21) exhibit significant weight gain, opposite to what is seen in HD patients (*Gray et al., 2008*; *Slow et al., 2003*; *Southwell et al., 2017*; *Southwell et al., 2013*). Another commonly used mouse model zQ175 KI has robust phenotypes (*Lin et al., 2001*); however, it has mouse HTT gene and therefore not suitable for testing the approach of genetically deleting human mutant HTT gene via methods such as CRISPR-Cas9.

Therefore, despite all the achievements, we are still in search for a new full-length model that not only accurately recapitulates all HD phenotypes, but also without spurious characteristics.

In light of the clear relationship between PolyQ repeat length and disease onset and severity in both humans and mice, in this study, we engineered a bacterial artificial chromosome (BAC) transgenic mouse model of HD-BAC226Q which expresses full-length human HTT with 226Q encoded by a mixture of CAG-CAA repeat. This 226Q repeat length is smaller than the maximum 250 polyQ in HD patient cases (*Nance et al., 1999*). The CAG-CAA mix will stabilize the polyQ length. Our novel BAC226Q mouse recapitulates a full spectrum of cardinal HD symptoms and pathologies: reduced life span, weight loss, motor and non-motor neurologic phenotypes, selective brain atrophy, striatal neuronal death, mHtt aggregation, and gliosis. It is also important to note that there are no spurious

abnormalities identified to date. Therefore, the new HD mouse model will be valuable for studying pathogenic mechanisms and developing therapeutics.

## Results

### Generation of BAC226Q transgenic mice

Transgenic BAC226Q mice were generated and contain the full-length human HTT with 226 CAG-CAA repeats under the control of the endogenous human HTT promoter and regulatory elements. Similar to the BACHD mouse, a mixed CAG-CAA repeat is used to create a stable length of polyglutamine (PolyQ) that is not susceptible to expansion or retraction (*Gray et al., 2008*). To determine the copy number and insertion sites in FVB mouse genome of full-length human HTT, whole-genome sequencing was applied and indicated two copies of the human HTT were inserted in the chromosome 8 in FVB mouse genome at Chr8:46084002 (*Figure 1A*). The protein expression level of HTT was determined by Western blots with whole-brain lysates from 2- to 11-month-old animals probed with anti-polyQ MAB1574 (1C2 clone) and S830 antibody against mutant huntingtin protein (*Figure 1B and C*). Age-matched 11-month BACHD and wild-type mice were used for comparison. As expected, the band for 226Q mutant huntingtin in BAC226Q mouse is at a higher molecular weight than that for 97Q huntingtin in the BACHD mouse. There are no Htt protein fragments detected in 2- or 11-month BAC226Q mouse in Western blots (*Figure 1B*).

### Progressive weight loss and shortened life span in BAC226Q mice

As a first step in characterizing these mice, their growth and reproduction were examined. Both male and female transgenic mice were born in the expected Mendelian ratios with no obvious abnormalities. Both males and females were fertile, but males were preferred for breeding because female animals had a short window of time during which they were able to adequately care for pups. Transgenic BAC226Q mice had significantly reduced lifespan (*Figure 2A*). In both sexes, only 50% of transgenic animals survive past 1 year. The longest surviving transgenic mice were 15 months old when they had to be euthanized due to the end-point condition according to animal welfare requirements. In comparison, non-transgenic littermate controls appeared healthy at 15 months of age. Animal weights were recorded weekly and the results show that HD mice were born normal and gained weight at the same rate as their non-transgenic littermates in the developmental stage (*Figure 2B*). In both male and female WT littermates, body weight continuously increased until the 11 months of age; however, BAC226Q exhibited slower weight gain followed by progressive and significant weight loss which resulted in about 40% loss at 11 months (*Figure 2C and D*). Weight loss is a common and serious complication in HD patients, who fail to maintain weight even when caloric intake is increased (*Aziz et al., 2008*). BAC226Q mice recapitulated this aspect accurately. These initial results suggest that BAC226Q mice have progressive weight loss and shortened lifespan, both of which correlate well with the human disease condition.

### Age-dependent and progressive motor impairments in BAC226Q mice

The earliest, most visible motor phenotype in the BAC226Q mice was the chorea-like movement which first manifested between 12 and 14 weeks. The motor phenotypes include abrupt unnatural jerking and twisting of the head and body (*Video 1*), resembling the chorea movement in HD patients. It should be noted that every transgenic BAC226Q mouse exhibits this HD-like movement. Between 14 and 16 weeks, rapid circling behavior appeared, and was especially prominent when animals were disturbed in their cages (*Video 2*).

To characterize the progressive nature of motor deficits in our model, we measured motor function at serial time points by several behavioral tests. First, overall activity was measured in the open field task (*Figure 3A and B*). Open field activity is divided into horizontal and vertical components. Horizontal activity is measured by lateral movement around the open field, and vertical activity is measured by rearing movements. At 2 months, HD mice are indistinguishable from non-transgenic littermate controls in both horizontal and vertical activity measures. Horizontal activity is dramatically increased at 4 months (*Figure 3A*), but returns to wild-type levels at 10 and 15 months. In contrast, vertical activity of 4-month HD mice has declined to about 20% of the controls (*Figure 3B*). This suggests that in addition to their hyperkinesia, the mice cannot rear normally, indicating a loss of

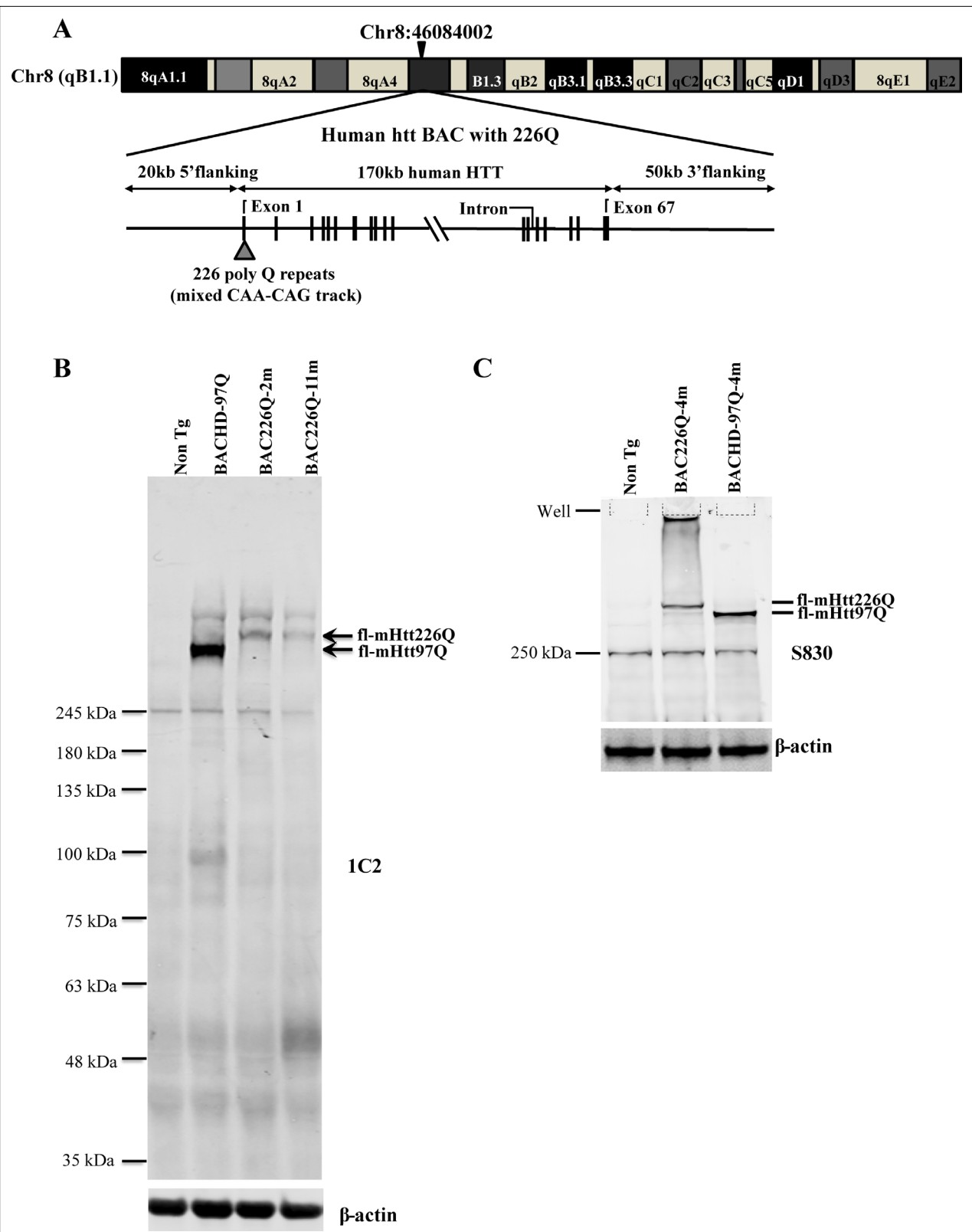

**Figure 1.** Generation of BAC226Q transgenic mice. (**A**) Schematic diagram of the insertion of the full-length human mutant HTT BAC into mouse genome. The human HTT BAC contains 170 kb genomic DNA of complete HTT genomic locus plus approximately 20 kb 5' and 50 kb 3' flanking region with endogenous regulatory elements. A mixed CAA-CAG repeat encoding 226 polyQ was engineered in exon 1. By whole-genome sequencing, two copies of the human mHTT BAC had been detected and inserted into the mouse genome at Chr8:46084002. (**B**) Western blot analysis of Htt protein

*Figure 1 continued on next page*

Figure 1 continued

expression levels in BAC226Q, non-transgenic littermate and BACHD (97Q) mice with antibody 1C2. The whole-brain lysates are from 2- to 11-month BAC226Q, 11-month non-transgenic littermate and BACHD mice. Western blots were repeated in three independent cohorts of mice. The upper panel: arrows indicate that 1C2, an antibody specific for mutant Htt, detects mHtt in BAC226Q and BACHD (97Q) mice but not non-transgenic littermate. In the second lane from the left, mHtt-97Q from BACHD (97Q) mouse appeared at the expected molecular weight. In the right two lanes, mHtt-226Q from 2- to 11-month BAC226Q mice are detected at a molecular weight higher than that of mHtt-97Q. Lower panel: the same blot was probed with anti-β-actin antibody for the loading control. (**C**) Reconfirmation by Western blot with the S830 antibody specific for mHtt in 4-month non-transgenic control, BAC226Q and BACHD (97Q) mice. S830 antibody detected mHtt-97Q in BACHD (97Q) mouse (right lane) and mHtt-226Q (middle lane) at expected molecular weights but not in non-transgenic control (left lane).

motor control. While reduced vertical activity at 4 months may be caused by the increase in horizontal movement, at 10 and 15 months, horizontal activity is reduced to wild-type levels but vertical activity does not recover and becomes even more impaired. These results parallel the biphasic motor symptom progression in HD patients in which early stage involuntary movements become more dystonic in late stages of disease.

The cylinder test was performed to provide another sensitive analysis of motor phenotypes. Similar to the open field test, BAC226Q mice are normal at 2 months and defective at 4 months in the cylinder test (*Video 3*). At 4 months, rearing frequency was unchanged, but rearing time was significantly reduced (*Figure 3C and D*). These results suggest that the transgenic mice retained their instinct to rear to explore the environment, but were unable to sustain upright posture due to motor impairment. The circling phenotype in BAC226Q mice (*Figure 3E*) was reminiscent of that seen in unilateral lesion models, but with important differences. In lesion models, stereotyped turning occurs in a single direction determined by which hemisphere is lesioned. While BAC226Q mice spent time turning in both directions, and the frequency of turning was significantly increased over controls. Circling in BAC226Q mice may be explained by a reduced capacity to inhibit or terminate movements such as turns.

To demonstrate deficits in balance and coordination and to track the progressive decline of motor function from normal at 2 months to severe at 4 months, an accelerating rotarod test was used. The accelerating rotarod requires mice to walk forward on a rotating rod to maintain balance and is a sensitive measurement of motor coordination. Consistent with the open field and cylinder results, no differences between BAC226Q and controls could be detected until 12 weeks (*Figure 3F*). At 13 weeks, BAC226Q mice performed significantly worse compared to non-transgenic controls. At 16 weeks, BAC226Q mice were unable to balance on the rotarod for any duration of time. These results demonstrated a rapidly progressing, debilitating motor dysfunction with a clearly defined time course.

## Cognitive and psychiatric-relevant deficits in BAC226Q mice prior to motor abnormalities

In HD patients, cognitive and psychiatric symptoms are common and as detrimental to quality of life as motor disorders. To characterize the non-motor phenotypes in the BAC226Q mice, we subjected 2-month-old BAC226Q mice to several tasks. Testing was performed at 2 months for two reasons: first, these tests depend to some degree on the ability of the animal to move around, thus we need to test at 2 months when there is no measurable motor impairment. Second, psychiatric and cognitive symptoms frequently occur years before the onset of motor symptoms in HD patients. The tests were performed in an ascending order of stress level, from object-in-place memory test, sucrose preference to the most stressful forced swim task. Because there is an evidence of impaired hippocampal function in pre-symptomatic HD patients, we chose a challenging cognitive task with a hippocampus-dependent spatial component (*Phillips et al., 2008*; *Ransome et al., 2012*). In this task, mice were exposed to four unique objects during a training session, and then exposed to the same four objects but with the positions of two objects switched (*Figure 4A*). Mice with normal cognitive function will recognize and spend more time exploring the switched objects. BAC226Q mice, on the other hand, showed no discrimination of the moved objects (*Figure 4A*) despite spent an equal total amount of time exploring the objects in both training and testing sessions. These results suggest that the BAC226Q mice have impairments in hippocampal-dependent cognitive function at 2 months.

Depression and anhedonia are also common in HD patients. To test if our mice displayed similar mood phenotypes, we applied three well-characterized behavioral assays. In the sucrose preference

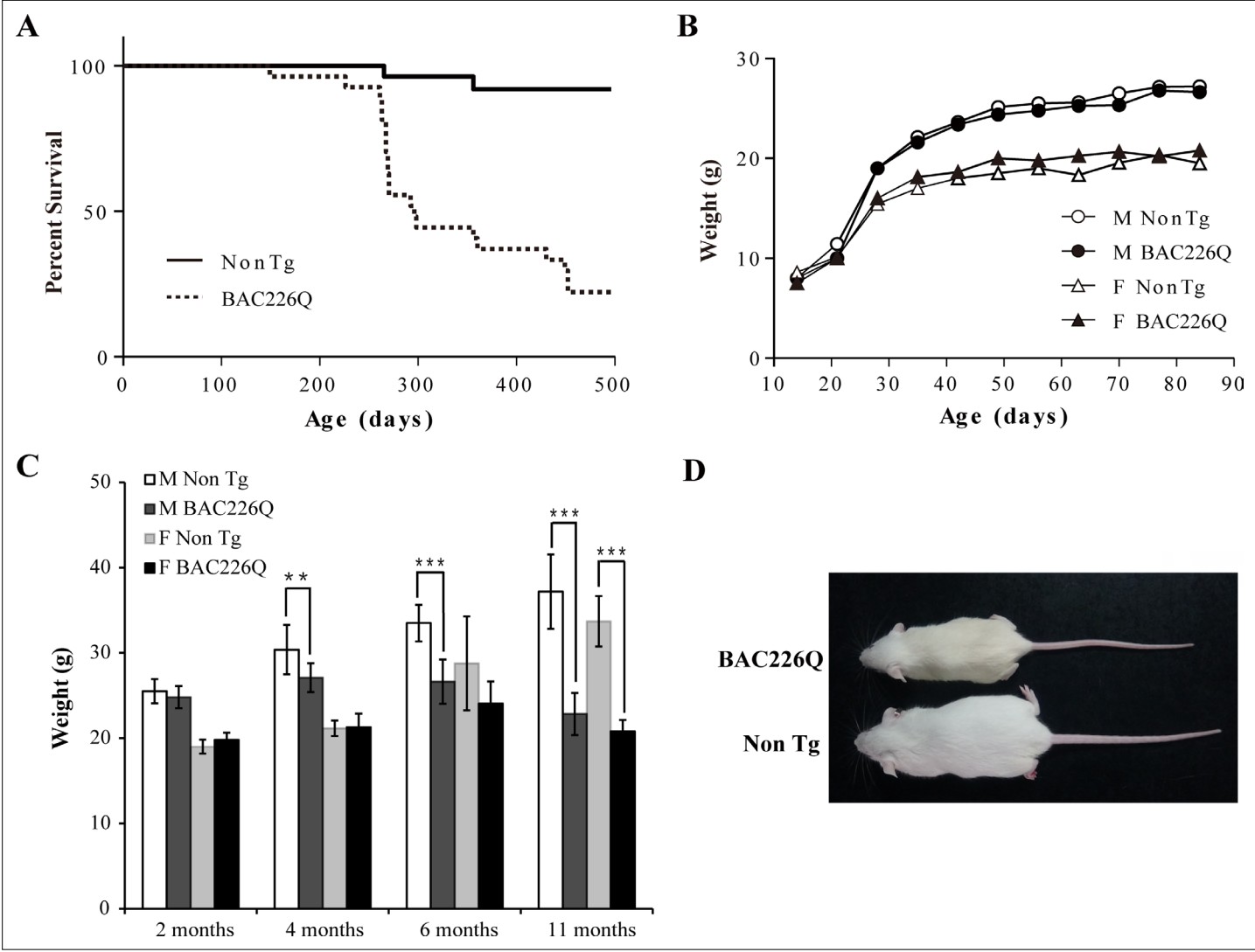

**Figure 2.** Shortened life span and weight loss in BAC226Q mice. (**A**) Kaplan-Meier survival curves indicate a median life span of less than 1 year for BAC226Q mice (male BAC226Q n=16; female BAC226Q n=11; male Non Tg n=19; female Non Tg n=19). (**B**) In the developmental stage, male and female BAC226Q mice gained weights at the same rate as their non-transgenic littermates (male BAC226Q n=8; male Non Tg n=10; female BAC226Q n=8; female Non Tg n=7). Fit a growth curve model to the mice, BAC226Q and non-transgenic littermates gained weights at the same rate (male 0.381±0.060 g/day, p=0.7123; female 0.156±0.035 g/day, p=0.09). (**C**) Progressive and age-dependent weight loss in BAC226Q mice. After normal development, BAC226Q mice had progressive weight loss compared to non-transgenic littermate controls: at 2 months (male BAC226Q n=8; male Non Tg n=10; p=0.3910; female BAC226Q n=8; female Non Tg n=7, p=0.1930); at 4 months (male BAC226Q n=10; male Non Tg n=13; p=0.0046; female BAC226Q n=8; female Non Tg n=6, p=0.8988); at 6 months (male BAC226Q n=22; male Non Tg n=20; p=1.6061e−11; female BAC226Q n=10; female Non Tg n=11, p=0.0884); at 11 months (male BAC226Q n=22; male Non Tg n=22; p=1.4608e−17; female BAC226Q n=6; female Non Tg n=8, p=1.9311e−06). Student's t-test was applied in all analyses. (**D**) Representative body sizes of 11-month-old male BAC226Q mouse and non-transgenic littermate. Significance is indicated by *=p<0.05, **=p<0.01, ***=p<0.001.

task, 2-month-old animals were given a choice of regular drinking water or 1% sucrose solution, and consumption of both was measured over 48 hr. While non-transgenic littermates preferred the sucrose solution, BAC226Q mice showed significantly reduced preference (*Figure 4B*), indicating an anhedonia-like phenotype typical of depression. The splash test is another test of depression-like mood dysfunction in mice. When sprayed with a sticky sucrose solution, BAC226Q mice spent significantly less time grooming compared to non-transgenic littermate (*Figure 4C*), demonstrating further a depression-like behavior. In the third experiment to explore depression-like phenotypes, we evaluated immobility in the forced swim task. The results were highly significant that BAC226Q mice had three times higher immobility time than non-transgenic littermate controls (*Figure 4D*). Combined,

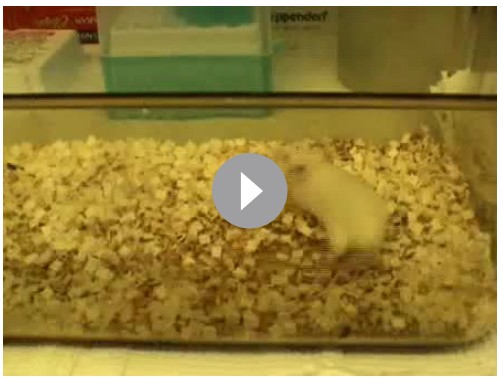

**Video 1.** Chorea-like movement in BAC226Q mice at 14 weeks.
https://elifesciences.org/articles/70217/figures#video1

these three tests indicated a range of depression-like phenotypes including disinterest in rewarding stimuli, apathy, and hopelessness in a stressful situation.

## Age-dependent and progressive striatal atrophy and neuronal loss in BAC226Q mice

Post-mortem brains of HD patients have a significant reduction in brain volume, especially in the striatum and deep layer cortex. Age-dependent and progressive striatal atrophy and neuronal loss are cardinal neuropathology of HD patients that are well recapitulated in BAC226Q mice (*Figure 5A*). At 2 months, BAC226Q mice had similar brain weights compared to non-transgenic littermates, and went through a progressive decrease. At 15 months, whole brains from surviving BAC226Q mice weighed 31% less than brains from non-transgenic littermate (*Figure 5B*). To further evaluate HD-like neuropathology in BAC226Q mice, we used the Cavalieri stereologic estimator to compare striatal volume in 11-month BAC226Q and non-transgenic littermates. The results showed that BAC226Q striatal volume was significantly decreased by 34% compared to controls (*Figure 5C*). Additionally, similar to the enlarged brain ventricles in HD patients due to brain atrophy, the volume of the ventricles in BAC226Q mice was greatly enlarged by 3.8-fold compared to that of the non-transgenic littermates (*Figure 5D*). These changes in striatal and ventricular volumes closely mirrored the cardinal pathology in HD patient brains.

In human HD patients, striatal volume loss is attributed largely to the death of MSNs, which comprise 95% of neurons in the striatum (*Oorschot, 1996*). To determine whether the striatal volume reduction in BAC226Q mice was caused by neuronal death, we used an unbiased optical fractionator method to count the total striatal neurons in brain tissues from 2- to 11-month animals. At 2 months, no significant differences were detected between genotypes with an average striatal neuron population of 2,280,000±125,000 for non-transgenic and 2,174,000±127,000 for HD mice (*Figure 5E*). However, at 11 months, non-transgenic littermate control mice had an average of 1,988,000±193,000 striatal neurons, while BAC226Q mice had only 1,631,000±269,000, a difference of 18.0% (*Figure 5E*). To further evaluate MSN health in striatum, DARPP-32 immunostaining was quantified by integrated optical density (IOD) in BAC226Q and non-transgenic littermate controls. DARPP-32 IOD was significantly reduced by 18.1% in BAC226Q striatum at 11 months of age (*Figure 5F*).

## Specific pattern of regional brain atrophy in BAC226Q mice by MRI study

To further characterize the degeneration in the HD mouse brain and to evaluate the feasibility of performing in vivo imaging to track disease progression, we imaged 12-month-old mice with high-resolution structural MRI and used automated deformation-based morphometry to determine regional brain volumes (*Figure 6A*). In 12-month transgenic BAC226Q mice, whole brain volume was 373.8±2.95 mm³, compared to 522.3±21.7 mm³ in non-transgenic controls, a decrease of 28.5% (*Figure 6B*). In order to make a fair comparison of regional differences between genotypes, we normalized all volumes by whole brain volume to obtain a percent volume for each region before performing statistical tests. We found statistically significant volume changes in several regions of the brain, including the cortex

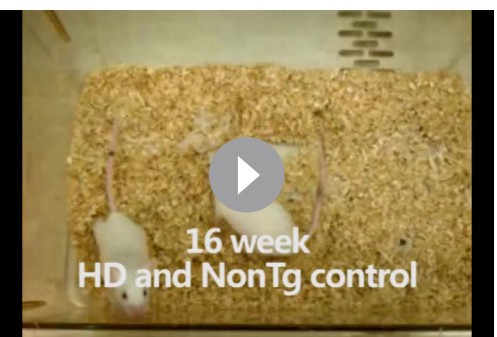

**Video 2.** Rapid circling behavior in BAC226Q mice at 16 weeks.
https://elifesciences.org/articles/70217/figures#video2

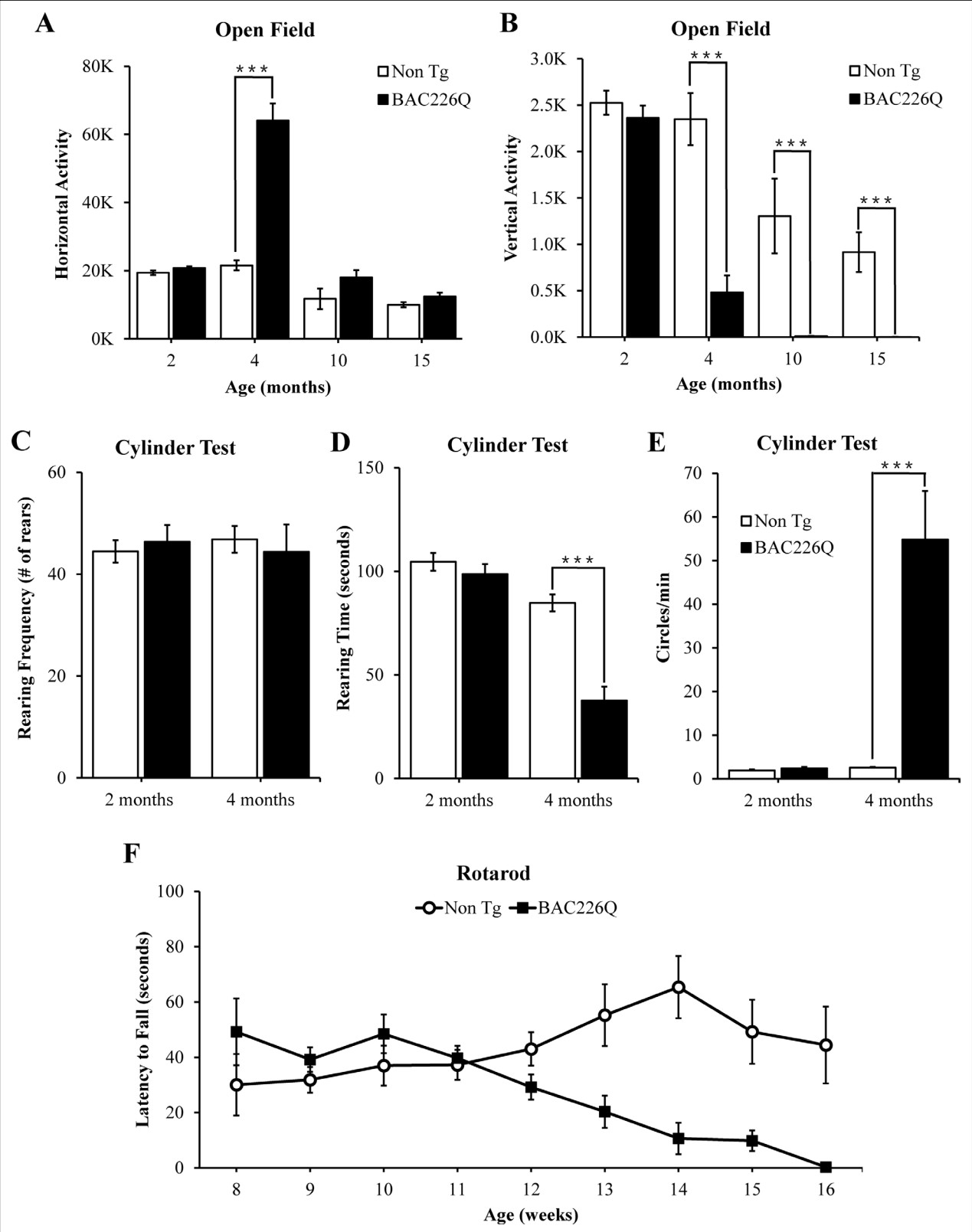

**Figure 3.** Robust, early onset, and progressive motor deficits in BAC226Q mice. (**A**, **B**) Horizontal and vertical activities are measured by total beam breaks in 1 hr in the open field. In horizontal movement, 4-month BAC226Q mice developed robust hyperactivity (p=3.7366e−06) (**A**). In vertical movement, BAC226Q mice had progressively diminished activity (2 months p=0.39450; 4 months p=0.00007; 10 months p=0.0015; 15 months p=0.00072) (**B**). (**C–E**) Results of the cylinder task at 2 and 4 months. Rearing frequency, the total number of rears observed during the 5-min task, is

*Figure 3 continued on next page*

Figure 3 continued

unchanged in BAC226Q mice (2 months p=0.3889; 4 months p=0.7016) (**C**). Rearing time, the total time of mice in an upright rearing position, is greatly reduced in BAC226Q mice at 4 months (p=0.00002) (**D**). Circling frequency, the total number of clockwise and counterclockwise rotations, is obviously increased in BAC226Q mice at 4 months (p=0.0013) (**E**). Student's t-test was applied in (**A–E**), and significance is indicated by ∗=p<0.05, ∗∗=p<0.01 and ∗∗∗=p<0.001. (**F**) Rotarod performance was averaged over three trials, performed once a week. BAC226Q mice showed progressive and significant deficits (13 weeks p=0.0004; 14 weeks p=0.0004; 15 weeks p<0.0001; 16 weeks p<0.0001); Two-way ANOVA with post hoc Sidak's multiple comparisons test, with a significant genotype and age interaction (F (8, 180)=5.805, p<0.0001). All tests used 11 pairs of BAC226Q mice and non-transgenic littermates.

(24.8%, p=0.0113, *Figure 6C*) and striatum (19.4%, p=0.0453, *Figure 6D*). There was a significant increase in the ventricle volume in the BAC226Q mice compared to non-transgenic littermate controls (206.8%, p<0.0001, *Figure 6E*), which was consistent with the Cavalieri stereologic estimator data (*Figure 5D*). In contrast, there was no significant difference in cerebellar and amygdala volumes between genotypes (*Figure 6F and G*). This is consistent with the observation that the cerebellum and amygdala are largely unaffected in HD patients. Other regions affected at 12 months included cingulum, stria medullaris, anterior commissure, and corpus callosum/external capsule, suggesting a pronounced effect on white matter tracts (*Figure 6H*). It is important to note that the detection of robust and progressive structural deficits as a biomarker by MRI provides a non-invasive and sensitive method for future therapeutic testing.

## Aggregate pathology and reactive gliosis in the BAC226Q brain

In post-mortem HD brains, mHtt aggregations are found throughout the central nervous system as an important hallmark (*Mangiarini et al., 1996*). To characterize the distribution patterns of huntingtin aggregations in the specific brain regions of BAC226Q mice, we stained brain tissues from 2-, 4-, and 11-month-old BAC226Q and non-transgenic littermate control mice with the S830 antibody raised against N-terminal huntingtin, which specifically detects soluble and aggregated mutant Htt (*Figure 7*). In agreement with previous findings (*Gray et al., 2008*; *Kazantsev et al., 1999*; *Li et al., 2001*), two types of huntingtin aggregates were observed in BAC226Q mice, nuclear inclusions (NIs) and neuropil aggregates (NAs).

At 2 months, neuron cell bodies were diffusely stained by the S830 antibody throughout the brain in BAC226Q mice (*Figure 7A–F*), but no immunoreactivity detected in non-transgenic littermates (*Figure 7S-X*). This suggests that at 2 months only soluble mHtt protein is present. In contrast, punctate staining of mHtt protein by S830 became evident in several regions of the brain at 4 months, including striatum (*Figure 7G*), motor (*Figure 7H*) and cingulate cortex (*Figure 7I*), hippocampus (*Figure 7J and K*) and amygdala (*Figure 7L*) in BAC226Q mice, but not control mice. At 11 months, aggregate pathology was much more severe in BAC226Q mice (*Figure 7M–R*).

Reactive gliosis is another cardinal pathology of HD in addition to selective neuronal loss. We analyzed reactive gliosis by immunohistochemical detection of GFAP, the astrocyte-specific marker, in BAC226Q mice and non-transgenic littermate. An 80% increase in reactive gliosis was found in deep layer cortex and striatum at 11-month BAC226Q mouse brain (*Figure 8*).

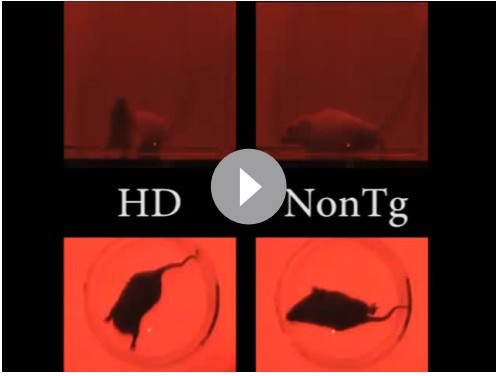

**Video 3.** Cylinder test in BAC226Q mice at 16 weeks.
https://elifesciences.org/articles/70217/figures#video3

## Discussion

Many mouse models have been developed for investigating the underlying pathogenic mechanisms and testing therapeutic methods. Categorically, a mouse model will be compelling if it meets two criteria simultaneously: accurate recapitulation of cardinal HD phenotypes in one mouse model rather than separately in several models, and absence of erroneous and unwanted false phenotypes that do not correlate with human HD. It should be emphasized that the novel BAC226Q reported here is such an example.

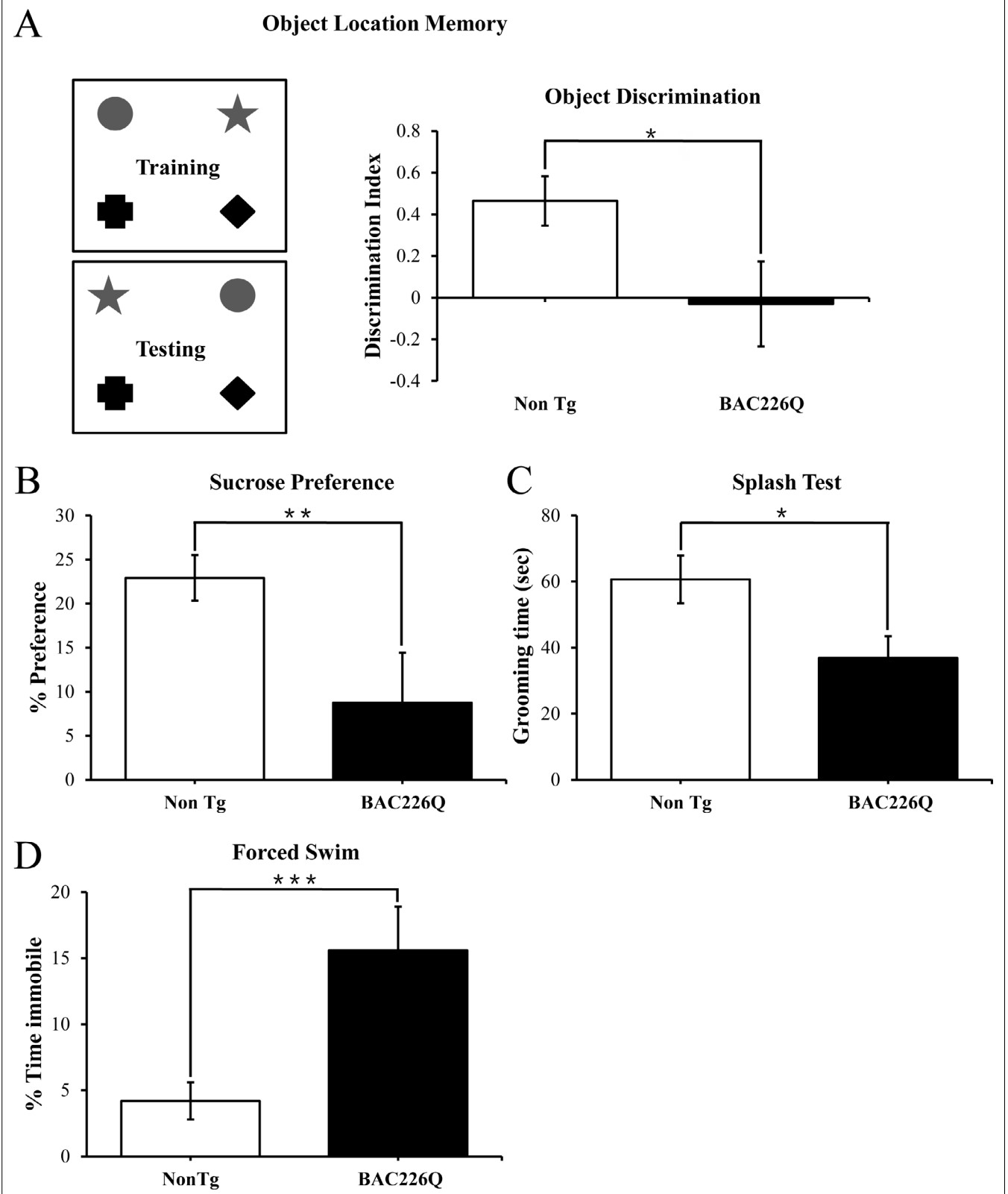

**Figure 4.** Cognitive and psychiatric disorder-like behavior in BAC226Q mice. All tests used 12 pairs of 2-month-old male BAC226Q mice and non-transgenic littermates. BAC226Q mice showed significant deficits in (**A**) significant decrease in discrimination of moved objects (p=0.0485), (**B**) significant decrease in sucrose preference (p=0.0031), (**C**) significant decrease in grooming time in the splash test (p=0.035), (**D**) significant increase in the immobile time in the forced swim test (p=0.0007). Student's t-test was applied in all analyses. Significance is indicated by *=p<0.05, **=p<0.01, ***=p<0.001.

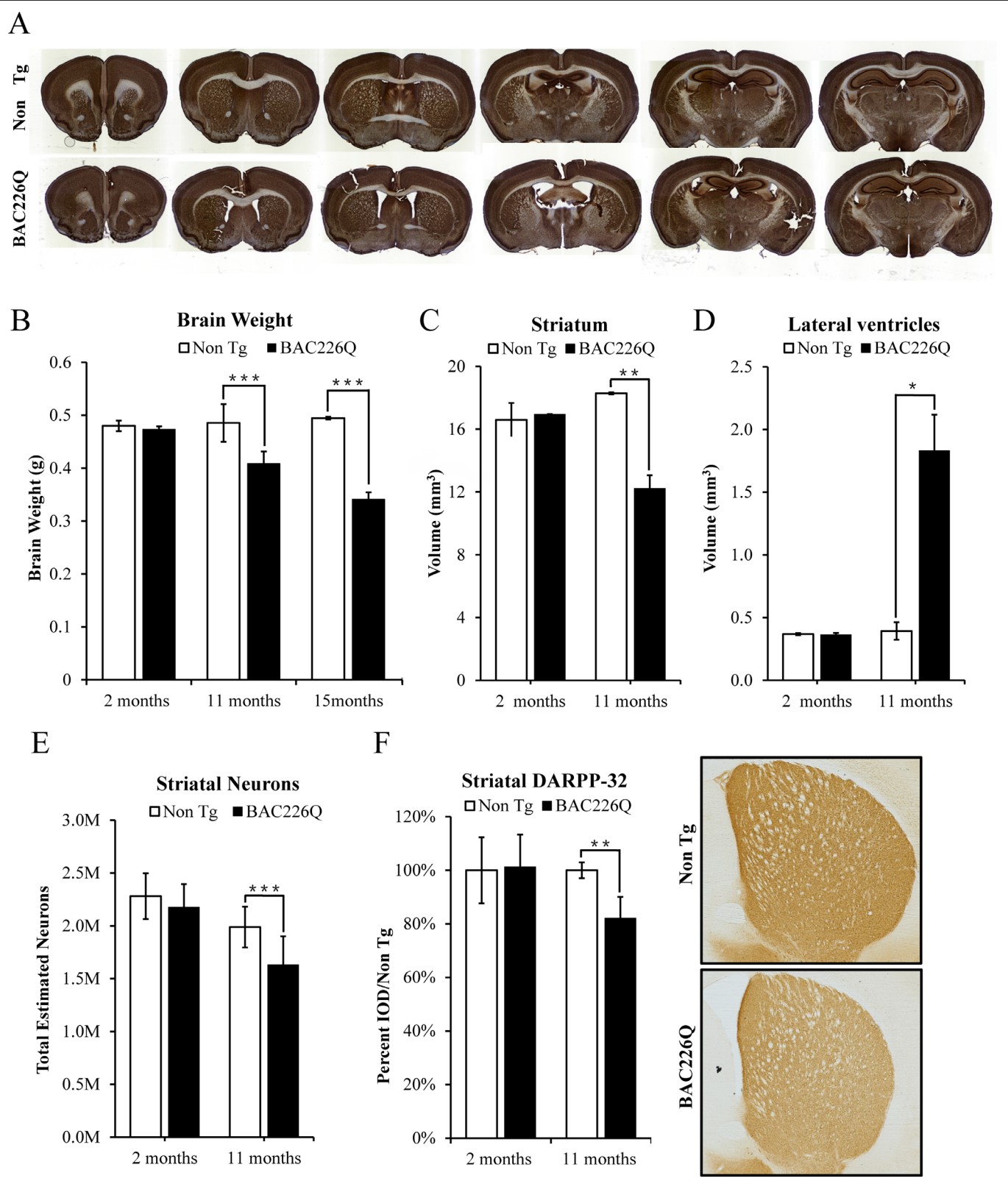

**Figure 5.** Histopathology of striatal atrophy and neuronal loss in BAC226Q mice. (**A**) Representative micrographs of NeuN stained serial brain sections for 11-month BAC226Q and non-transgenic littermates. (**B**) Progressive and age-dependent brain weight loss in BAC226Q mice. With the normal development, brain weights of BAC226Q mice is similar to non-transgenic littermates at 2 months (BAC226Q 0.473±0.005 g; Non Tg 0.480±0.010 g, n=3 pairs; p=0.373901). Brain weights of BAC226Q mice were significantly reduced at 11 months (BAC226Q 0.408±0.023 g, Non Tg 0.485±0.035 g, reduced

*Figure 5 continued on next page*

*Figure 5 continued*

by 15.9%; BAC226Q n=9, Non Tg n=15; p=0.00026), and more severe at 15 months (BAC226Q 0.495±0.013 g, Non Tg 0.341±0.007 g, reduced by 31.0%; n=4 pairs; p=0.00019). (**C**, **D**) Striatal atrophy was measured by the Cavalieri stereological estimator. Compared to the non-transgenic littermates, BAC226Q at 2 months had no defects in striatal (BAC226Q 16.556±0.007 mm³, Non Tg 16.167±1.091 mm³; n=3 pairs; p=0.84283) and ventricle volume (BAC226Q 0.345±0.014 mm³, Non Tg 0.351±0.007 mm³; n=3 pairs; p=0.81590). At 11 months, BAC226Q striatal volume was significantly decreased by 34% (BAC226Q 11.789±0.829 mm³, Non Tg 17.765±0.130 mm³; n=4 pairs; p=0.00544) (**C**), and lateral ventricle volume was increased by 379% (BAC226Q 1.765±0.281 mm³, Non Tg 0.369±0.067 mm³; n=4 pairs; p=0.01270) (**D**). (**E**) Total striatal neuron count was estimated by an optical dissector stereological estimator. No significant difference was detected at 2 months (BAC226Q 2,280,000±125,000, Non Tg 2,174,000±127,000; n=3 pairs), but a significant 18% decrease was detected in 11-month BAC226Q (BAC226Q 1,631,000±269,000,, Non Tg 1,988,000±193,000; BAC226Q n=6, Non Tg n=5, p=0.0002). In every subject counted, the estimated coefficient of error was less than 0.1. (**F**) In 11-month BAC226Q mice striatum, DARPP-32 staining of MSNs was reduced by 18.1% (n=4 pairs, p=0.0012) but no change in 2-month BAC226Q mice striatum. In all panels, Student's t-test was applied in all analyses. Significance is indicated by ∗=p<0.05, ∗∗=p<0.01 and ∗∗∗=p<0.001.

## Robust and faithful recapitulation of HD

As presented in the Results section and *Table 1*, the BAC226Q mouse model has validity and fidelity in a full spectrum from genomic DNA, protein, subcellular/cellular pathology, histopathology, specific brain area atrophy, cognitive and psychiatric disorder-like phenotypes, motor behavioral deficits, weight loss, and shortened life spans.

At the genomic level, the human BAC clone contains the complete human HTT gene with its endogenous intron-exon structures and 5′, 3′ regulatory regions. This is important for testing future gene editing approach such as CRISPR/Cas9-mediated targeting. BAC226Q transgenic mice express full-length human HTT with 226 CAG-CAA repeats, which yielded robust and early onset HD phenotypes. It should be noted that 226Q is within the range of mutations identified in human patients (*Nance et al., 1999*). In additional consideration for practical usage, it is worth noting that the CAG-CAA mixture gives BAC226Q mice an important advantage of complete polyQ length stability between generations and individuals.

Importantly, BAC226Q mice do not have spurious phenotypes that sometimes exist in other models. For example, unlike other models with severe developmental problems, BAC226Q mice develop normally and subsequently have age-dependent and progressive neurodegeneration. Also, unlike other models that showed significant body weight gain which is opposite to human HD, BAC226Q mice have progressive weight loss and reduced lifespan, consistent with human HD weight loss as a hallmark (*Gaba et al., 2005*; *Stoy and McKay, 2000*). Interestingly, since weight loss is also observed in BAC-225Q model with full-length mouse mutant HTT (*Van Raamsdonk et al., 2006*; *Wegrzynowicz et al., 2015*), it seems that weight loss phenotype is polyQ-length dependent rather than human or mouse HTT species dependent.

Compared to other human full-length HTT models, a significant advantage of BAC226Q mice is its much earlier onset, much more robust, and faster progressing phenotypes including early hyperkinetic/late hypokinetic biphasic motor dysfunction. HD patients develop hyperkinetic chorea first and as the disease progresses, hypokinesia and dystonia subsequently. Parallel to the patient's motor symptom progression, BAC226Q mice have severe hyperkinesia including involuntary choreiform movement by 12–16 weeks and show reduced mobility after 10 months.

An important but less well-studied aspect of HD is psychiatric and cognitive impairments, which often occur decades before the onset of motor symptoms in HD mutation carriers. In addition to motor dysfunctions, BAC226Q mice show non-motor phenotypes at 2 months, before the onset of motor deficit and neuropathology, which is the same temporal sequence as in patients. Thus BAC226Q is an appropriate and powerful tool to study the mechanisms underlying psychiatric and cognitive deficits.

## A model well suited for preclinical investigations

Several features make BAC226Q well suited for preclinical studies. First, BAC226Q accurately recapitulates HD phenotypes. Therefore, candidate drugs or approaches for disease modification can be tested for their abilities to rescue these highly relevant phenotypes at multiple levels. Second, BAC226Q has normal development and very early onset of phenotypes that are progressive and robust. This provides a long window of observation for testing the efficacy of therapeutic candidates. Third, the remarkable region-specific brain atrophy revealed by high-resolution structural MRI can be used as a biomarker and readout in preclinical studies longitudinally without sacrificing the animals.

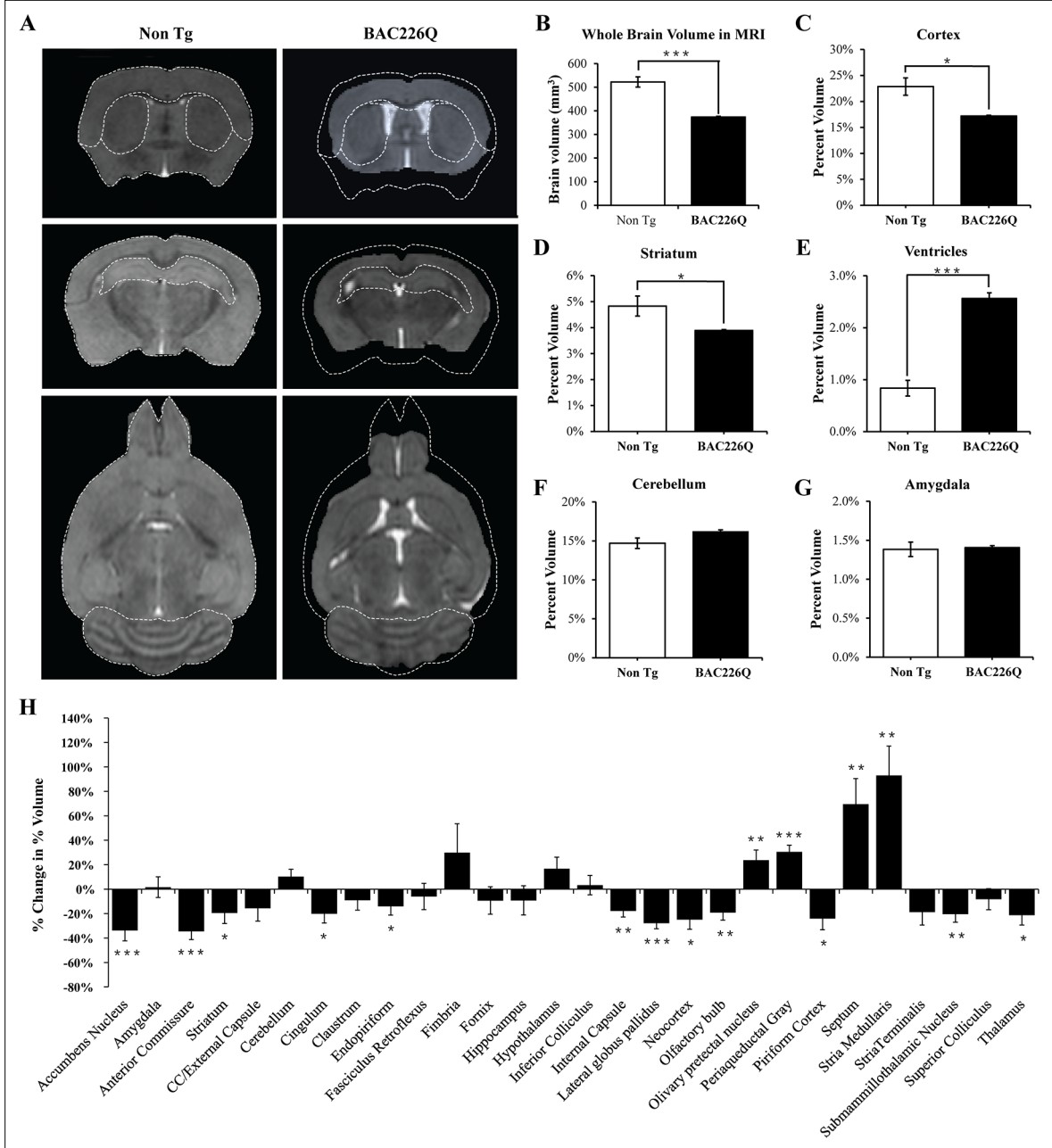

**Figure 6.** MRI study of regional specific brain atrophy in BAC226Q mice. (**A**) MRI images of 12-month female BAC226Q and non-transgenic littermates. (**B–G**) Overall brain volumes were segmented automatically into regional brain volumes by MRI volumetric analysis in BAC226Q mice (n=9) and non-transgenic littermates (n=6). The volumes are presented as the percentage of whole brain volume. BAC226Q mice have significant decreases in (**B**) whole brain volumes (28.5%, p=0.00022), (**C**) relative cortex volume (24.8%, p=0.0113) and (**D**) striatum (19.4%, p=0.0453), and significant increase in ventricle volume (**E**) (206.8%, p=0.0001). There is no significant difference between genotypes in (**F**) cerebellar (p=0.0618) and (**G**) amygdala volume (p=0.8229). (**H**) Total 28 brain regions were measured by MRI volumetric analysis. The changes of specific brain region volume in BAC226Q mice are presented as the percentage of the same brain area in non-transgenic littermates. In BAC226Q mice, 16 brain regions had significant volume changes (p<0.05). Student's t-test was applied in all analyses. Significance is indicated by ∗=p<0.05, ∗∗=p<0.01 and ∗∗∗=p<0.001.

Fourth, the CAG-CAA mix makes the polyQ length stable between generations and among individuals, which is a critical advantage of consistency for research and especially for drug development, although this model is limited for research in somatic instability and RAN translation, two of the actively studied mechanisms that are suggested to be important for HD pathogenesis (*Bañez-Coronel et al., 2015*; *Grima et al., 2017*; *Kacher et al., 2021*; *Swami et al., 2009*; *Tabrizi et al., 2020*). Last

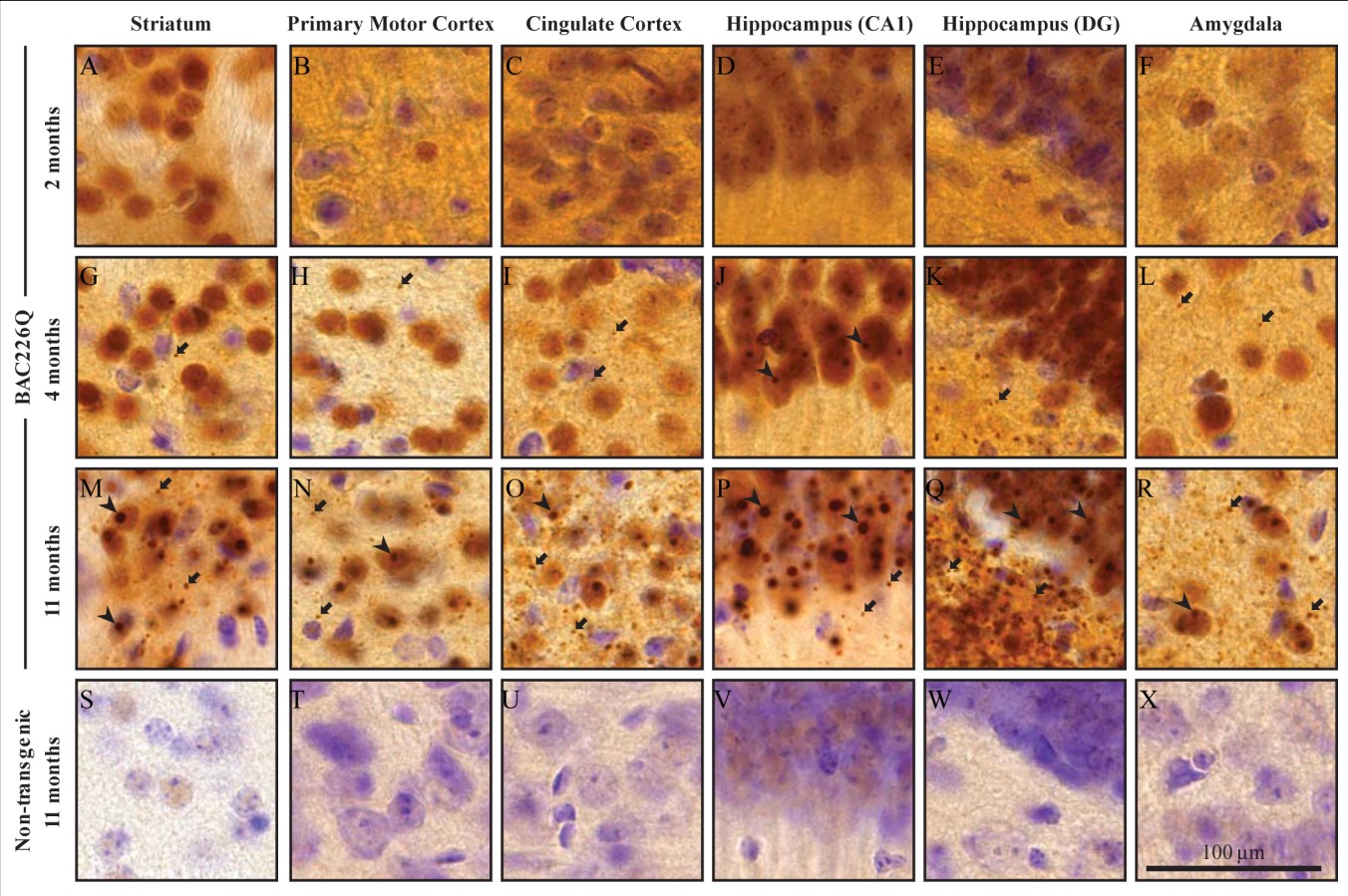

**Figure 7.** Widespread and progressive mHtt aggregate pathology in BAC226Q brain. Huntingtin aggregates were stained by the S830 antibody. In BAC226Q, mHtt aggregates were undetectable at 2 months (**A–F**), detected as small aggregates (arrows) and nuclear inclusions (arrowheads) at 4 months (**G–L**), finally developed to large and numerous cytosolic aggregates and nuclear inclusions at 11 months (**M–R**). In contrast, no aggregates were detected in non-transgenic littermates (**S–X**). Regions shown are striatum (**A, G, M, S**), primary motor cortex (**B, H, N, T**), cingulate cortex (**C, I, O, U**), CA1 of hippocampus (**D, J, P, V**), dentate gyrus (**E, K, Q, W**), and amygdala (**F, L, R, X**). Scale bar, 100 µm.

but not least, BAC226Q has human genomic DNA as the transgene, which is very appropriate for testing gene-editing such as CRISPR/Cas9 strategy in preclinical studies.

## Insights of mHtt toxicity in BAC226Q

Although we only conducted an initial analysis of BAC226Q mice, we have already generated data that can be used to clarify some of the questions in the field.

Huntingtin aggregates were first identified in R6 mice expressing exon 1 of mutant huntingtin and subsequently identified in human tissue (*Mangiarini et al., 1996*). Since N-terminal huntingtin is much more susceptible to aggregation compared to the full-length protein (*Ratovitski et al., 2009*), and causes a more severe disease phenotype in mice, there has been a hypothesis that full-length mHtt need to be cleaved into fragments to exert toxicity. Although further investigation is needed to detect various fragments, our repeated Western blots of full-size gels for different ages of BAC226Q, exemplified by *Figure 1B*, have not shown appreciable 226 PolyQ-containing N-terminal fragments nor their accumulation as a function of time and disease stages in BAC226Q mice. It will be interesting to explore in the future whether the low abundance N-terminal fragments below detection on Western blots are the main cause of toxicity, or the full-length mHtt with 226Q played the main role in pathogenesis in this mouse model.

Another important question is whether mHtt has a dominant gain-of-toxic function, or a combination of loss-of-function of the wild-type allele. In human patients, rare cases of disrupting one Htt allele did not develop abnormality (*Ambrose et al., 1994*), patients with homozygous mutant Htt alleles

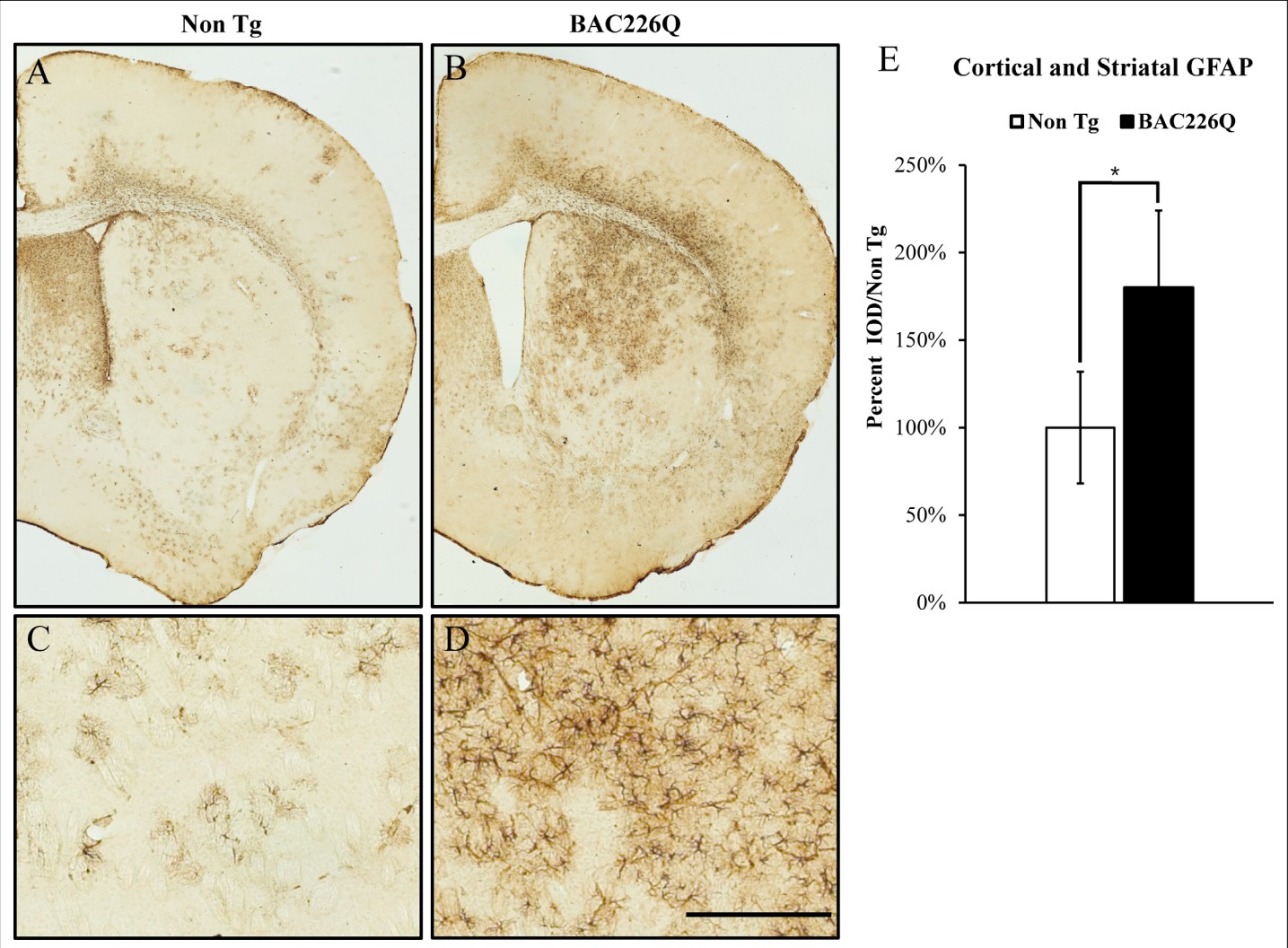

**Figure 8.** Gliosis in striatum and deep cortical layers of BAC226Q brain. Reactive astrogliosis is analyzed by immunohistochemical staining with GFAP antibody in 11-month BAC226Q and non-transgenic littermate brains. Gliosis is prominent in BAC226Q (**B, D**), but not in control littermate brains (**A, C**) (n=4, p=0.032, Student's t-test). Scale bar, 100 µm.

develop HD similar to heterozygous carriers (*Dürr et al., 1999*; *Kremer et al., 1994*; *Squitieri et al., 2003*; *Wexler et al., 1987*). In BAC226Q, mHTT was not overexpressed, and the two alleles of the wild-type HTT exist. The fact that BAC226Q mice developed such robust HD-like phenotype and that wild-type Htt knockout in adult mice did not induce HD-like phenotypes (*Leavitt et al., 2020*; *Wang et al., 2016*), seem to give more support to the 'gain-of-function' hypothesis. An interesting future experiment will be to put BAC226Q in Htt$^{+/-}$ and Htt$^{-/-}$ background, and examine whether the current HD-like phenotypes in BAC226Q are enhanced. The implication of the above discussion is whether deleting mutant HTT allele will be sufficient to benefit patients in gene-targeting as a therapeutic method.

## Conclusions

In this study, we report the generation and analyses of a novel BAC226Q mouse, which accurately recapitulates the cardinal HD phenotypes including body weight loss, HD-like characteristic motor behavioral impairment, cognitive and psychiatric symptoms, and classic HD neuropathology changes such as significant neuronal death in striatum and cortex, widespread mHtt aggregation pathology, and reactive gliosis. Therefore, this model will be valuable for mechanistic studies and therapeutic development of HD, especially for the preclinical genetic therapies targeting human mHTT.

**Table 1.** Recapitulation of cardinal HD phenotypes in BAC226Q mice.

| HD patients | | BAC226Q mice |
|---|---|---|
| Neuropathology | Brain atrophy | Yes |
| | Neuron loss | Yes |
| | mHTT aggregations | Yes |
| | Reactive gliosis | Yes |
| Progressive motor deficits | Chorea | Yes |
| | Incoordination | Yes |
| | Dystonia | Yes |
| | Bradykinesia | Yes |
| Non-motor symptoms | Psychiatric symptoms | Yes |
| | Cognitive deficits | Yes |
| Reduced life span | | Yes |
| Weight loss | | Yes |

## Materials and methods

### BAC engineering and generation of transgenic mice

A BAC containing the full-length wild-type human HTT gene was modified to express a full-length HTT with a stable, expanded polyglutamine tract. A plasmid containing exon 1 of HTT with 226 mixed CAG-CAA repeats was a gift from Dr. Alex Kazantsev (MIT). The sequence corresponding to 226Q was inserted into the HTT gene by homologous recombination using our standard protocols (*Gong et al., 2002*). Fingerprinting analysis by genomic Southern blots detected no unwanted rearrangements or deletions in the modified Htt-226Q BAC. The modified BAC was sequenced to confirm that there were no unwanted mutations other than the intended 226Q insertion. Finally, the full-length human Htt-226Q BAC copy number and the insertion site in mouse genome were determined by whole-genome sequencing and bioinformatics analysis, performed by Novogene Corporation.

### Animal husbandry

Mice were housed in a temperature and humidity controlled specific pathogen-free (SPF) facility under a 12-hr light/dark cycle schedule with food and water available ad libitum. Transgenic mice were bred with wild-type FVB/N mice (Taconic). It is important to note that BAC226Q in the Taconic FVB/N mouse background exhibited the most robust and consistent phenotypes. Genotyping was determined by PCR of genomic DNA isolated from tail snips. Genotyping primers are located in intron 2 and are specific to human huntingtin (forward primer: 5'-GTA TAT GCT GCT GCC TGC AA-3'; reverse primer: 5'-AGG GGA CAG TGT TGG TCA AG-3'), producing a 403-bp PCR fragment. Non-transgenic littermates were used as controls. All experimental protocols were approved by the Weill Cornell Medicine and Peking University animal care and use committee.

### Motor behavioral study

All motor behavioral tests were performed during the dark phase of the light-dark cycle for male and female mice. Data were counted by investigators who were blind to the genotypes of mice.

#### Open field test

Mice were monitored individually with the VersaMax Animal Activity Monitoring System (Accuscan Instruments). The activity monitor consists of a 16× 16 grid of infra-red beams at floor level to track horizontal position with 16 additional beams at a height of 3 in. to detect vertical activity. Male and female animals were used in test at the ages of 2, 4, 10, and 15 months. Horizontal and vertical activities were measured by beam breaks recorded over a period of 1 hr by the VersaMax software in the dark cycle. The open field chambers were thoroughly cleaned between animals to ensure removal of odors from previous animals. Data were recorded and analyzed by the VersaMax software (Accuscan Instruments). There was no significant influence of sex, thus male and female data were pooled for analysis.

#### Cylinder test

Male and female animals were used in test at the ages of 2 and 4 months. Each animal was placed in a clear acrylic cylinder and recorded on video for 5 min. A mirror was placed below the cylinder to provide a view of the animal from below. For rearing frequency, rearing duration, and circling behavior, video recordings were analyzed using ImageJ. Digital videos of the mice were converted

to gray-scale and cropped to a view of the animal's silhouette. With VirtualDub software, a Gaussian blur filter was applied and followed by thresholding to obtain a clear outline of the animal's body. The videos were subsequently analyzed with the 'Analyze Particles' tool in ImageJ to obtain pixel area and orientation angle for each frame (5 min at 30 frames per second). Animal rotation was determined by measuring changes in orientations, that is, the angle of the major axis of the best-fit ellipse between frames in either clockwise or counterclockwise direction. The degree of continuous rotations above 180° in either direction were summed and divided by 360° to obtain the number of circling. All three sets of data were calculated for each individual animal by ImageJ, and confirmed by manual counts. There was no significant influence of sex, thus male and female data were pooled for analysis.

## Rotarod

Mice were trained to run on the rotarod apparatus (IITC Life Science) at 8 weeks of age in three daily trials at a fixed speed (10 rpm) for 60 s. The training lasted for 3 days. Only during this training period, mice who fell were placed back on the rod until the end of the 60 s. In subsequent experiments, mice were tested once a week in a session of three consecutive trials on an accelerating rotarod (4–45 rpm over 5 min). The interval between trials is 15 min. Latency to fall was recorded for each animal and averaged over the three trials. The same cohort of wild-type and BAC226Q mice were used for this experiment.

## Cognitive and psychiatric behavioral study

The same cohort of male mice for cognitive and psychiatric abnormalities were singled housed for 3 days prior to and during object location memory, sucrose preference, splash, and forced swimming tests. These mice were not re-used for motor behavioral tests.

## Object location memory

Mice were introduced to an open field chamber for 5 min for 3 days. On day 4, four unique objects were placed to the open field chamber. On day 5, mice were presented with the same four objects but the position of two objects was switched. Discrimination Index (DI) was calculated using the formula $DI = \frac{T_m - T_u}{T_m + T_u}$ where $T_m$ and $T_u$ are time spent exploring the moved and unmoved objects, respectively.

## Sucrose preference test

Individually housed mice were trained to drink from two identical water bottles on day 0. On day 1, one water bottle was replaced with a 1% sucrose solution and on day 2, the positions of bottles were switched. The volumes of plain water and sucrose solution before and after experiments were recorded. Preference Index was calculated by the following formula, where $C_S$ and $C_W$ are sucrose consumed and water consumed, respectively:

$$\%\text{preference} = \frac{C_S - C_W}{C_S + C_W} \times 100\%$$

## Splash test

Mice were sprayed on their dorsal coat with a 10% sucrose solution, placed in an empty cage, and recorded for 5 min. Duration of grooming activity was scored by a person blind to genotype.

## Forced swim task

Mice were placed individually into a cylinder of room temperature water (25±1°C) and recorded for 5 min. Immobility (time spent floating and not swimming) was scored manually by a person blind to genotype.

## Western blot

Freshly dissected mouse brains were homogenized by a Precellys tissue homogenizer in ice-cold RIPA buffer (150 mM NaCl, 0.1% sodium dodecyl sulfate (SDS), 1% Sodium deoxycholate, 50 mM Triethanolamine, 1% NP-40, and pH 7.4) supplemented with Complete Protease Inhibitor Cocktail tables (Roche). The supernatant was collected after centrifugation in a refrigerated centrifuge (4°C) at 16,000×*g* for 20 min. Protein concentration was determined by a BCA Assay (Pierce). An equal amount of protein (120 µg/sample) were denatured in 4× LDS sample buffer (141 mM Tris base,

106 mM Tris HCl, 2% LDS, 0.51 mM EDTA, 10% glycerol, 0.22 mM SERVA Blue G-250, 0.175 mM phenol red, and pH 8.5) with 10× reducing buffer or 4× SDS sample buffer (0.2 mol/L Tris-HCl pH 6.8, 0.4 mol/L DTT, 8% SDS, 0.4% bromophenol blue, and 40% glycerol) and incubated at 95°C for 10 min. Protein samples were loaded in each well of NuPAGE 4–15% Tris-Acetate pre-cast gradient gels (Invitrogen) or 15×22 cm², 8% polyacrylamide gels for SDS-PAGE. Proteins were transferred to a PVDF membrane (Immobilon-FL) and blocked with LI-COR Odyssey blocking buffer, then probed with an expanded polyQ specific antibody 1C2 (MAB1574, Millipore, 1:5000) or sheep antibody S830 against N-terminal mHtt (a gift from Dr. Gillian Bates, 1:20,000) or rabbit anti-tubulin (Chemicon, 1:1000). Blots were incubated with IRDye conjugated secondary antibodies (1:10,000) then visualized by an Odyssey Infra-red Imaging System (LI-COR Biosciences).

## Neuropathology

### Tissue preparation

Mice were anesthetized with a ketamine/xylazine cocktail and transcardially perfused with PBS followed by 4% freshly prepared, ice-cold paraformaldehyde (PFA). Brains were removed and post fixed for 24 hr in PFA, then transferred into PBS for storage. Brains were mounted in 4% agarose and coronal sections were obtained with a vibratome (40 μm and 60 μm thickness). Serial sections were collected into 10 wells so that each well contained every 10th section. Floating sections were stored in anti-freeze buffer (30% glycerol and 30% ethylene glycol in PBS) at –20°C.

### Immunostaining

Floating sections were stained with the NeuN antibody (Millipore, 1:1000), the S830 anti-Htt antibody (a gift from Dr. Gillian Bates, 1:25,000), the anti-GFAP antibody (Abcam, 1:2000), and the DARPP-32 antibody (Abcam, 1:7500) for overnight at 4°C. Subsequent incubations in secondary antibody, ABC solution, and color development were performed according to instructions provided with the ABC and DAB substrate kits (Vector Labs). Brain sections stained by NeuN antibody and S830 anti-Htt antibody were counter-stained with cresyl violet.

### Stereology

Stereologic measurements were obtained by Stereo Investigator (MBF Biosciences) with a Zeiss Axio-phot2 microscope. The optical fractionator probe was used to estimate total striatal neuron number in every 10th section. The counting frame was 30× 30 s in the NeuN antibody stained brain sections with an optical dissector depth of 8 or 18 μm for 40 and 60 μm cut sections, respectively with 2 μm guard zones. At least 300 neurons were counted for each animal, and the estimated coefficient of error in each case was less than 0.1. Striatal and ventricle volumes were determined using the Cavalieri estimator on the same slides with a grid size of 100×100 μm² oriented at a random angle.

## MRI volummetry study

MRI scanning and computational analysis were performed as previously described (*Cheng et al., 2011*). Female BAC226Q and wild-type FVB mice aged 12 months were scanned by a 9.4 T MR scanner with a triple-axis gradient and animal imaging probe. The scanning was performed in vivo under isoflurane anesthesia. The resulting images were aligned to a template by automatic registration software and signal values were normalized to ensure consistent intensity histograms. A computer cluster running Large Deformation Diffeomorphic Metric Mapping (LDDMM) was used to automatically construct non-linear transformations to match anatomical features and perform automatic segmentation of specific brain regions. The volume of each segmented structure was normalized by total brain volume and analyzed for each genotype.

## Statistics

Data were shown as mean ± SEM (standard error of the mean) unless otherwise noted. Student's t-test (unpaired) or two-way ANOVA with post hoc Sidak's multiple comparisons test was used to compare transgenic and control groups in each experiment with significance level indicated by p-values. The means, SEM and p-values were calculated by Microsoft Excel 2010 and GraphPad Prism 8 software. Survival curves were compared with the log-rank (Mantel-Cox) test. Fit a growth curve model was applied in *Figure 2B*.

## Acknowledgements

This project was partially supported by Hereditary Disease Foundation, Weill Cornell Medical College, and Peking University School of Life Sciences. The authors wish to thank the National Center for Protein Sciences at Peking University for technical assistance. The authors declare no competing financial interests.

## Additional information

### Funding

| Funder | Grant reference number | Author |
|---|---|---|
| Hereditary Disease Foundation | | Chenjian Li |
| Weill Cornell Graduate School of Medical Sciences | | Chenjian Li |
| Peking University | | Chenjian Li |

The funders had no role in study design, data collection and interpretation, or the decision to submit the work for publication.

### Author contributions

Sushila A Shenoy, Conceptualization, Data curation, Formal analysis, Validation, Visualization, Writing – original draft, Writing – review and editing; Sushuang Zheng, Conceptualization, Data curation, Formal analysis, Investigation, Validation, Visualization, Writing – original draft, Writing – review and editing; Wencheng Liu, Data curation, Investigation, Methodology, Writing – review and editing; Yuanyi Dai, Yuanxiu Liu, Data curation, Formal analysis, Writing – review and editing; Zhipeng Hou, Susumu Mori, Data curation, Formal analysis; Yi Tang, Data curation, Methodology, Writing – review and editing; Jerry Cheng, Data curation, Methodology, Software, Validation, Writing – review and editing; Wenzhen Duan, Data curation, Formal analysis, Methodology, Writing – review and editing; Chenjian Li, Conceptualization, Data curation, Formal analysis, Funding acquisition, Investigation, Methodology, Project administration, Supervision, Validation, Writing – original draft, Writing – review and editing

### Author ORCIDs

Chenjian Li (iD) http://orcid.org/0000-0001-7905-6434

### Ethics

This study was performed in strict accordance with the recommendations in the Guide for the Care and Use of Laboratory Animals of Weill Cornell Medicine and Peking University. All animal studies were conducted in accordance with the Guide for the Care and Use of Laboratory Animals (8th edition) and approved by the Institutional Animal Care and Use Committee of Peking University. The laboratory approval number from the Association for Assessment and Accreditation of Laboratory Animal Care (AAALAC)-approved Animal Facility at Peking University Laboratory Animal Center (LAC-PKU) IACUC (#CFLS-LiCJ-1). All surgery was performed under sodium pentobarbital anesthesia, and every effort was made to minimize suffering.

### Decision letter and Author response

Decision letter https://doi.org/10.7554/eLife.70217.sa1
Author response https://doi.org/10.7554/eLife.70217.sa2

## Additional files

### Supplementary files

- Transparent reporting form

## Data availability

Source data has been deposited to Dryad (doi:https://doi.org/10.5061/dryad.qrfj6q5g0).

The following dataset was generated:

| Author(s) | Year | Dataset title | Dataset URL | Database and Identifier |
|---|---|---|---|---|
| Li C | 2022 | Data from: A novel and accurate full-length HTT mouse model for Huntington's disease | http://dx.doi.org/10.5061/dryad.qrfj6q5g0 | Dryad Digital Repository, 10.5061/dryad.qrfj6q5g0 |

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
