## [Editor Report]

This work describes creation of a novel Huntington disease (HD) mouse model that has potential value to the HD field. This model demonstrates disease features that have previously been difficult to capture and opens new avenues of examination.

---

## [Decision Letter]

**Decision letter after peer review:**

Thank you for submitting your article "A novel and accurate full-length HTT mouse model for Huntington's disease" for consideration by *eLife*. Your article has been reviewed by 3 peer reviewers, including Harry T Orr as the Reviewing Editor and Reviewer #1, and the evaluation has been overseen by Matt Kaeberlein as the Senior Editor. The following individual involved in review of your submission has agreed to reveal their identity: Amber L Southwell (Reviewer #2).

Essential revisions:

1. Given the potential importance of somatic instability in the HD pathogenesis as well as ongoing therapeutic strategies it is critical that authors discuss the absence of somatic instability in this model as a weakness.

2. If the authors are going to conclude that HTT fragments are not generated and thus do not contribute to pathogenesis in this model, it is critical that whether fragments are in fact generated be determined a method designed specifically to detect small proteins/peptides.

3. Statistical analysis is insufficient and not detailed. The statistical analysis section says that analyses were performed using the students t test or one way ANOVA, though which test was used for each piece of data is not detailed anywhere, and what tests were used for post hoc correction of multiple comparisons is not mentioned at all. In either the Results section or the figure legends, please indicate what test is used for each piece of data and give the exact p values for comparisons that are significant and those that are close to allow the reader to make their own assessments. Perhaps a statistical table could be included with the test, the n and the p for ANOVA as well as each pairwise comparison.

Additionally, in many cases, one way ANOVA is not appropriate. In cases where mice are assessed at multiple ages, 2WAY ANOVA for age and genotype must be employed. In cases where the same mice are used at multiple time points, this must be repeated measures ANOVA. Also, assessments at multiple ages performed using the same mice should be presented as line graphs, such as panel 2B, while those with different mice should be presented as bar graphs, such as 2C. If the bodyweight data in 2C was actually performed using the same mice at each age, it should not be presented as a bar graph.

Additional Points:

1. The assessment of HTT level and species should include WT Htt. Even though this is the mouse Htt protein, if it is not included, then an assessment of transgene expression level in comparison to endogenous expression cannot be made. From the Westerns that compare mHTt to BACHD mice, it appears that expression is lower. This is great since the BACHD is an overexpression model and a model with expression closer to endogenous levels would be a good thing. However, without the wtHtt, this assessment can't be made. Re-probing with MAB2166 would show both proteins, and considering the large size of the tg mHTT, separation of the proteins should not be a problem. It also looks from the S830 blot that there is a portion of the tg mHTT that remained insoluble. The authors may want to consider a more robust detergent in lysis in order to more accurately quantify tg mHTT.

2. Methods are insufficiently detailed. Were the same mice used for every study? If so, they would have been moved to single housing early on for the sucrose preference test. Since single housing affects other behavioral outcomes, and could do so in a disease-dependent manner, it would be important to understand this. An overview paragraph or diagram describing the cohorts is needed, and the potential effects of single housing on subsequent assessments should be addressed.

– Sex of the mice is only mentioned for body weight and for MRI. Were equal numbers of male and female mice used for other assessments? Was data analyzed separately for males and females and only combined if they were not different? Why were only female mice used for MRI?

– The authors observed and quantified circling. They correctly make the important distinction that the circling phenotype of mice with unilateral motor damage is unidirectional and state that the circling in these mice is bidirectional. However, in the videos, the mice seems to only circle in one direction. To make the statement that these mice circle bidirectionally, this analysis would need to be done separately for each direction.

– Neuropathology was assessed at 2 and 11 months of age, but data is only shown for the stereology at 2 months of age. The authors state in results that forebrain weight was similar at 2 months of age, but the data is not shown. In fact, the figure only shows whole brain weight at 11 months, not forebrain weight. Please describe how the forebrain was divided and add the data. Was the DARPP-32 assessment performed at 2 months? If not, why? If so, where is the data?

– The methods state that behavioral videos were analyzed with ImageJ, but no other information is given. Most labs use expensive animal tracking software to analyze behavior videos. This statement requires either a full description or a citation of another paper with a full description of exactly house mouse behavior was analyzed by ImageJ.

– Most of the behavior data has insufficient detail. How many training trials were done over how many days for rotarod? The performance of the WT mice seems awfully poor, but the accelerating program described is pretty standard except for a shorter than usual inter trial interval. My WT mice can stay on this program for basically the full 300s, yet the mean reported here is ~30s for young WT mice. Were they only trained 1 day? Considering that all of the FL tg HD mice have reported motor learning deficits in rotarod as early as 2months of age where it takes 3 days for them to catch up to WT mouse performance, but they can then perform just as well as the WT mice, this is really insufficient. Obviously it can't be repeated, but it needs to be clarified. The difference in performance between what is reported here and what is typically reported for FVB mice needs to be addressed. This 10 fold difference in performance is huge.

– Dilutions of antibodies should be given, and secondary antibodies should be named.

– Amount of protein separated on APGE should be given (only says 'an equal amount of protein').

3. Other conclusions are not supported by the data. The authors describe S830 as aggregate-specific in 1 sentence and say that it is picking up only soluble mHTT in the next paragraph. These are contradictory statements. S830 does, in fact, detect soluble mHTT, so should not be used as a basis for making statements about the role of aggregates in HD pathogenesis. The authors state that it is widely postulated that Htt aggregates are the source of HD toxicity. However, it has been well described that neurons with aggregates live longer than neurons without aggregates, suggesting a protective role of aggregation. Additionally, multiple mouse models with robust brain aggregation, shuch as the YAC128 C6R and short stop models, have transgene alterations that make them non-pathogenic i.e. no neurodegeneration, not HD-like behavior phenotype, but still result in robust brain HTT aggregates. So, it's been well established in the field for a long time that aggregation is not the driver of disease in HD.

– The authors state that because they observe robust disease in their model and the full complement of mouse Htt is still present, this demonstrates that HD is a GOF disease, not LOF. First, this is not a novel finding. Like several of the things in the Discussion section, it is common to other FL mouse models of HD, thus, not novel. This includes things like onset of cognitive and psychiatric behavioral abnormalities prior to motor abnormalities (Seen in YAC128, BACHD, Humanized mice, Q175), or robust disease without overt dependence on cleavage (hard to tell without fragment analysis, but presumably not different from BACHD). Furthermore, mHTT is known to sequester wtHTT or do its job poorly and potentially competitively, so just because the wtHTT is still there, doesn't' meant that the mHTT can't have some dominant negative effect and result in LOF toxicities. Thus, this study clarifies GOF vs LOF.

4. Often the figure call outs are out of order in the text. For instance, in figure 2, survival data is presented first followed by body weight data, whereas in the text, body weight data is presented prior to survival data. Please re-order either the text or the figure. This also occurs with panel 5A coming after 5B, and 6E coming after 6G in the text.

5. The introduction states that the causative gene for HD was identified 2 decades ago. This was actually 28 years; closer to 3 decades.

6. In the description of existing mouse model of HD, no distinction is made between KI models where only additional CAG is knocked in to the mouse Htt gene and models where the entirety of human HTT exon 1 is knocked in to the mouse gene. This is an important distinction for assessment of therapies that target exon 1 outside of the CAG tract.

7. Please include N in figure legends 3 and 4 as are done in 2 and 5.

8. One ref in the introduction is denoted as 4 author last names et al., while all others are only 1 name et al., or 2 names with and please change for consistency.

9. Weight loss Results section says that the tg mice gained weight the same as their littermates for the first 8 weeks, but the graph in 2B shows total overlap for 13 weeks (90 days).

10. DARPP-32 staining is not a way to quantify MSNs. Please rephrase this in results. It is a marker of healthy MSNs. Expressions can be reduced and lost prior to death of cells. Even if this wasn't the case, using IOD of staining throughout the striatum doesn't quantify cells.

---

## [Author Response]

Essential revisions:1. Given the potential importance of somatic instability in the HD pathogenesis as well as ongoing therapeutic strategies it is critical that authors discuss the absence of somatic instability in this model as a weakness.

The reviewer’s point is well taken. We acknowledge this limitation and revised in the Discussion:

“… Fourth, the CAG-CAA mix makes the polyQ length stable between generations and among individuals, which is a critical advantage of consistency for research and especially for drug development, although this model is limited for research in somatic instability and RAN translation, two of the actively studied mechanisms that are suggested to be important for HD pathogenesis (Bañez-Coronel et al., 2015; Grima et al., 2017; Kacher et al., 2021; Swami et al., 2009; Tabrizi et al., 2020)…”

2. If the authors are going to conclude that HTT fragments are not generated and thus do not contribute to pathogenesis in this model, it is critical that whether fragments are in fact generated be determined a method designed specifically to detect small proteins/peptides.

We thank the reviewer for this comment. We are fully aware of the body of publications on various mouse models, pro-and-con on this issue, including the significant ones of which the reviewer was a co-author. For example, some of the publications proposing the “fragment pathology” are (Graham et al., 2006; Slow et al., 2005). And the publications inconsistent with the above are (Gafni et al., 2012). In our case, we have done a large number of Western blots for brain samples at different ages in BAC226Q, and obtained consistent results that BAC226Q mice didn’t show detectable mHtt fragmentation. Being aware that more detailed investigation will be needed, we have softened our language in this revised version, making it a “tentative discussion” rather than a “conclusion”.

Our previous manuscript reads:

“Our data do not support this hypothesis, because in the life span of BAC226Q mice, fragmented mHtt is hardly detectable. It seems that in BAC226Q, full length mHtt is toxic and sufficient to drive pathogenesis.”

Now we change the paragraph to:

“Although further investigation is needed to detect various fragments, our repeated Western blots of full-size gels for different ages of BAC226Q, exemplified by Figure 1B, have not shown appreciable 226 PolyQ-containing N-terminal fragments nor their accumulation as a function of time and disease stages in BAC226Q mice. It will be interesting to explore in the future whether the low abundance N-terminal fragments below detection on Western blots are the main cause of toxicity, or the full length mHtt with 226Q played the main role in pathogenesis in this mouse model.”

1. Gafni, J., Papanikolaou, T., Degiacomo, F., Holcomb, J., Chen, S., Menalled, L., Kudwa, A., Fitzpatrick, J., Miller, S., Ramboz, S., et al. (2012). Caspase-6 activity in a BACHD mouse modulates steady-state levels of mutant huntingtin protein but is not necessary for production of a 586 amino acid proteolytic fragment. J Neurosci *32*, 7454-7465. 10.1523/jneurosci.6379-11.2012.

2. Graham, R.K., Deng, Y., Slow, E.J., Haigh, B., Bissada, N., Lu, G., Pearson, J., Shehadeh, J., Bertram, L., Murphy, Z., et al. (2006). Cleavage at the caspase-6 site is required for neuronal dysfunction and degeneration due to mutant huntingtin. Cell *125*, 1179-1191. 10.1016/j.cell.2006.04.026.

3. Slow, E.J., Graham, R.K., Osmand, A.P., Devon, R.S., Lu, G., Deng, Y., Pearson, J., Vaid, K., Bissada, N., Wetzel, R., et al. (2005). Absence of behavioral abnormalities and neurodegeneration in vivo despite widespread neuronal huntingtin inclusions. Proc Natl Acad Sci U S A *102*, 11402-11407. 10.1073/pnas.0503634102.

3. Statistical analysis is insufficient and not detailed. The statistical analysis section says that analyses were performed using the students t test or one way ANOVA, though which test was used for each piece of data is not detailed anywhere, and what tests were used for post hoc correction of multiple comparisons is not mentioned at all. In either the Results section or the figure legends, please indicate what test is used for each piece of data and give the exact p values for comparisons that are significant and those that are close to allow the reader to make their own assessments. Perhaps a statistical table could be included with the test, the n and the p for ANOVA as well as each pairwise comparison.Additionally, in many cases, one way ANOVA is not appropriate. In cases where mice are assessed at multiple ages, 2WAY ANOVA for age and genotype must be employed. In cases where the same mice are used at multiple time points, this must be repeated measures ANOVA. Also, assessments at multiple ages performed using the same mice should be presented as line graphs, such as panel 2B, while those with different mice should be presented as bar graphs, such as 2C. If the bodyweight data in 2C was actually performed using the same mice at each age, it should not be presented as a bar graph.

We thank the reviewers for this critique. We first invited Prof. Jerry Cheng, a professional Bio-statistician to join our team as a co-author. Prof. Cheng worked with us in actually re-analyzing all data sets. As a result, we revised many places in *Figures* (substantial revision in Figure 2 and Figure 5), *Figure legends* (substantial revision in Figure 2 and Figure 5) and *Methods*. We have given more detailed explanations of exact P values, numbers of animals in each experiment, and the types of statistical analyses. It is worth noting that regarding some of the repeated measurements, e.g., Figure 2C, we made serious efforts to use exactly the same cohort of mice in longitudinal studies. However, a complication is that as the mice aged, many of them, especially BAC226Q mice, died in the process. Therefore, we were forced to add additional mice from the same breeding batch but not in the original experimental cohort to reach sufficient numbers of animals. Thus, these experiments were not strict repeated measures. We explained each individual case in the revised Figure Legends and Methods.

Additional Points:1. The assessment of HTT level and species should include WT Htt. Even though this is the mouse Htt protein, if it is not included, then an assessment of transgene expression level in comparison to endogenous expression cannot be made. From the Westerns that compare mHTt to BACHD mice, it appears that expression is lower. This is great since the BACHD is an overexpression model and a model with expression closer to endogenous levels would be a good thing. However, without the wtHtt, this assessment can't be made. Re-probing with MAB2166 would show both proteins, and considering the large size of the tg mHTT, separation of the proteins should not be a problem. It also looks from the S830 blot that there is a portion of the tg mHTT that remained insoluble. The authors may want to consider a more robust detergent in lysis in order to more accurately quantify tg mHTT.

We thank the reviewer for pointing this out. In fact, in the past 5 months, this is the experiment in which we spent most of our effort and struggled with severe obstacles including obtaining antibodies from international shipping to China. We tried multiple antibodies, many conditions of lysis buffers, and different types of gels. The 1C2 antibody doesn’t recognize the wild type Htt. We tried MAB2166 in 10 gels and Western blots, just to confirm a problem well known in the field that MAB2166 has much reduced affinity to mHtt in the reverse relationship to the length of expanded PolyQ. BACHD97 was poorly recognized by MAB2166, and BAC-225Q was almost not recognized (Wegrzynowicz et al., 2015). BAC226Q is similar to BAC-225Q. Therefore, our tentative estimate of BAC226Q expression level comes from the 1C2 and S830 detection of relative abundance of mHtt226Q and mHtt97Q which is known to be 2-3 folds of the wild type Htt.

We will pursue another approach that can potentially address this question with Evotec SE Company by the MSD method (Reindl *et al.*, 2019), and will eventually give a more definitive answer. At this moment, however, we would like to ask your permission to publish with our current data. We revised the text to:

“Without definitive quantification, the mHtt226Q expression level was estimated by its relative signal to that of mHtt97Q detected by 1C2 and S830 antibodies. More accurate quantification will be performed with MSD method (Evotec SE Company).”

For your reference but not for addition in the manuscript, Author response images 1 and 1 is to demonstrate our effort in clarifying this issue, by examples of two sets Western blot analyses of Htt protein expression levels in BAC226Q, non-transgenic littermate and BACHD (97Q) mice with antibody MAB2166.

**Author response image 1. sa2fig1:** 

1. Wegrzynowicz, M., Bichell, T.J., Soares, B.D., Loth, M.K., McGlothan, J.S., Mori, S., Alikhan, F.S., Hua, K., Coughlin, J.M., Holt, H.K., et al. (2015). Novel BAC Mouse Model of Huntington's Disease with 225 CAG Repeats Exhibits an Early Widespread and Stable Degenerative Phenotype. J Huntingtons Dis *4*, 17-36.

2. Reindl, W., Baldo, B., Schulz, J., Janack, I., Lindner, I., Kleinschmidt, M., Sedaghat, Y., Thiede, C., Tillack, K., Schmidt, C., et al. (2019). Meso scale discovery-based assays for the detection of aggregated huntingtin. PLoS One *14*, e0213521. 10.1371/journal.pone.0213521.

Results:Western blot indicates that MAB2166 detects very well endogenous wildtype mouse Htt in BAC226Q, BACHD (97Q) mice and non-transgenic littermates, with reduced affinity to mHtt in BACHD (97Q), and not in BAC226Q (226Q).

A) The whole-brain lysates are from 2- and 11-month BAC226Q, 11-month non-transgenic littermate and BACHD mice. Protein samples are mixed with 4xLDS sample buffer and loaded with NuPAGE 4-12% Bis-Tris gel. The upper arrow indicates that MAB2166 only can detect mHtt in BACHD (97Q) mice at the expected molecular weight, but neither in BAC226Q mice nor in non-transgenic littermate. The lower arrow indicates that MAB2166 detects endogenous mouse Htt in BAC226Q and BACHD (97Q) mice as well as in non-transgenic littermate.

B) The whole-brain lysates are from 2- and 11- month BAC226Q, 11-month non-transgenic littermate and BACHD mice and loaded with 15 x 22 cm, 8% polyacrylamide gels. Similarly with panel A), MAB2166 detects endogenous wildtype mouse Htt in BAC226Q, BACHD (97Q) and non-transgenic littermate as indicated in the lower arrow, and only detects mHtt in BACHD (97Q) mice indicated in the upper arrow in the second lane but not in BAC226Q mice.

Methods:

Freshly dissected mouse brains were homogenized by a Precellys tissue homogenizer on ice-cold RIPA buffer (150mM NaCl, 0.1% sodium dodecylsulfate (SDS), 1% Sodium deoxycholate, 50mM Triethanolamine, 1% NP-40, pH 7.4) supplemented with Complete Protease Inhibitor Cocktail tables (Roche). The lysates were incubated at 4°C for 1 h and the supernatants were collected after centrifugation in a refrigerated centrifuge at 16,000 xg for 20 minutes. Protein concentration was determined by a BCA Assay (Pierce), and protein samples were denatured in 4xSDS sample buffer (0.2 mol/L Tris-HCl pH 6.8, 0.4 mol/L DTT, 8% SDS, 0.4% bromophenol blue, 40% glycerol) or 4xLDS sample buffer (141 mM Tris base, 106 mM Tris HCl, 2% LDS, 0.51 mM EDTA, 10% glycerol, 0.22 mM SERVA Blue G-250, 0.175 mM phenol red, pH 8.5) with 10x reducing buffer and incubated at 95°C for 10 minutes. An equal amount of protein (120µg) was loaded in each well of NuPAGE 4-12% Bis-Tris gel (Invitrogen), or 15 x 22 cm 8% polyacrylamide gel for SDS-PAGE. Proteins were transferred to a PVDF membrane (Immobilon-FL) and probed with MAB2166 (Millipore, 1:1000) in Odyssey Blocking Buffer. IRDye 680RD Goat anti-Mouse IgG was used as secondary antibody (Li-COR, 1:10000). An odyssey infra-red imager was used to visualize the blots (Li-COR biosciences).

2. Methods are insufficiently detailed. Were the same mice used for every study? If so, they would have been moved to single housing early on for the sucrose preference test. Since single housing affects other behavioral outcomes, and could do so in a disease-dependent manner, it would be important to understand this. An overview paragraph or diagram describing the cohorts is needed, and the potential effects of single housing on subsequent assessments should be addressed.– Sex of the mice is only mentioned for body weight and for MRI. Were equal numbers of male and female mice used for other assessments? Was data analyzed separately for males and females and only combined if they were not different? Why were only female mice used for MRI?

We added the following explanation in Figure Legends and Methods:

We tried to keep consistency within an experiment with regard to using males or females. For all cognitive and psychiatric behavioral tasks, we used males only to avoid estrus cycle complication. The same cohort of male mice for cognitive and psychiatric abnormalities were singled housed for 3 days prior to and during object placement, sucrose preference, splash and forced swimming tests. These mice were not re-used for motor behavioral tests.

In studies of motor deficits, we compared males and females and didn’t find a significant difference in open field and cylinder tests. In rotarod test, we fit a regression model with the average latency-to-fall as the outcome and age-in-weeks as the covariate. Male and female BAC226Q mice showed progressive and strong deficits (males from 12 weeks, p < 0.001; female mice from 10 weeks, p <0.001). Furthermore, we combined males and females in two-way ANOVA analysis with Sidak post hoc test, and found a dominant and significant genotype and age interaction (F (8, 180) =5.805，p<0.0001).

In preliminary studies of pathology, we didn’t find a sex specific difference. Subsequently, histopathology was performed with mixed sexes and MRI study with female mice.

– The authors observed and quantified circling. They correctly make the important distinction that the circling phenotype of mice with unilateral motor damage is unidirectional and state that the circling in these mice is bidirectional. However, in the videos, the mice seems to only circle in one direction. To make the statement that these mice circle bidirectionally, this analysis would need to be done separately for each direction.

All mice have bi-directional circling. Interestingly, in one session a mouse tends to circle mainly to one direction, and when it is picked up and put back to the cage, the mouse usually circles in random choice of either direction.

– Neuropathology was assessed at 2 and 11 months of age, but data is only shown for the stereology at 2 months of age. The authors state in results that forebrain weight was similar at 2 months of age, but the data is not shown. In fact, the figure only shows whole brain weight at 11 months, not forebrain weight. Please describe how the forebrain was divided and add the data. Was the DARPP-32 assessment performed at 2 months? If not, why? If so, where is the data?

We thank the reviewer for the suggestions. Now in Figure 5B, we revised to include data for 2, 11, and 15 months; in Figure 5F, we now include data for 2 and 11 months.

– The methods state that behavioral videos were analyzed with ImageJ, but no other information is given. Most labs use expensive animal tracking software to analyze behavior videos. This statement requires either a full description or a citation of another paper with a full description of exactly house mouse behavior was analyzed by ImageJ.

We added the following details in the *Method* section:

In the Open field Test, mice were monitored individually for 10 minutes in the VersaMax Animal Activity Monitoring System and video recordings were analyzed by the VersaMax software (Accuscan Instruments). Horizontal movements and vertical rearing were counted by investigators who were blind to the genotypes of the mice.

The cylinder test was performed during the dark phase of the light-dark cycle. Each animal was placed in a clear acrylic cylinder and recorded on video for 5 minutes. A mirror was placed below the cylinder to provide a view of the animal from below. Rearing frequency and time were counted from the video recording by investigators who were blind to the genotypes of the mice. For circling behavior, video recordings were analyzed using ImageJ. Digital videos of the mice in cylinder task were converted to gray-scale and cropped to a view of the animal’s silhouette. With VirtualDub software, a Gaussian blur filter was applied and followed by thresholding to obtain a clear outline of the animal’s body. The videos were subsequently analyzed with the “Analyze Particles” tool in ImageJ to obtain pixel area and orientation angle for each frame (5 minutes at 30 frames per second). Frequency and duration of rearing were calculated for each individual animal, and confirmed by manual counts by investigators who were blind to the genotypes of the mice. Animal rotation was determined by measuring changes in orientations, i.e., the angle of the major axis of the best-fit ellipse between frames in either clockwise or counter-clockwise direction. The degree of continuous rotations above 180 ^o^ in either direction were summed and divided by 360 to obtain the number of circling.

– Most of the behavior data has insufficient detail. How many training trials were done over how many days for rotarod? The performance of the WT mice seems awfully poor, but the accelerating program described is pretty standard except for a shorter than usual inter trial interval. My WT mice can stay on this program for basically the full 300s, yet the mean reported here is ~30s for young WT mice. Were they only trained 1 day? Considering that all of the FL tg HD mice have reported motor learning deficits in rotarod as early as 2months of age where it takes 3 days for them to catch up to WT mouse performance, but they can then perform just as well as the WT mice, this is really insufficient. Obviously it can't be repeated, but it needs to be clarified. The difference in performance between what is reported here and what is typically reported for FVB mice needs to be addressed. This 10 fold difference in performance is huge.

We have revised the Method for rotarod test as follows:

“Mice were trained to run on the rotarod apparatus (IITC Life Science) at 8 weeks of age in three daily trials at a fixed speed (10 rpm) for 60 seconds. The training lasted for three days. Only during this training period, mice that fell were placed back on the rod until the end of the 60 seconds. In subsequent experiments, mice were tested once a week in a session of three consecutive trials on an accelerating rotarod (4-45 rpm over 5 minutes). The interval between trials is 15 minutes. Latency to fall was recorded for each animal and averaged over the three trials. The same cohort of wild type and BAC226Q mice were used for this experiment.”

With regard to the reviewer’s question about the shorter than usual latency before falling for young wild type and BAC226Q mice, we think that the main factor is the difference in the equipment *per se*. When we performed the task of Figure 3F in New York, we used *IITC Life Science* rotarod, which perhaps was much more slippery than usual, and caused all animals to stay much shorter time on the rotating rod. Subsequently, Li Lab moved to Peking University and has been using *Rota-Rod MED ENV510 (MED Associates, Inc)*, and discovered that wild type mice consistently stayed on the rod with duration comparable to 300s.

– Dilutions of antibodies should be given, and secondary antibodies should be named.

We have revised and indicated all the detailed information in Methods:

“probed with an expanded polyQ specific antibody 1C2 (MAB1574, Millipore, 1:5000) or sheep antibody S830 against N-terminal mHtt (a gift from Dr. Gillian Bates, 1:20000) or rabbit anti-tubulin (Chemicon, 1:1000). Blots were incubated with IRDye conjugated secondary antibodies (1:10000)”

– Amount of protein separated on APGE should be given (only says 'an equal amount of protein').

We have added this information (120 µg per protein sample) in the Western Blot section of Methods.

3. Other conclusions are not supported by the data. The authors describe S830 as aggregate-specific in 1 sentence and say that it is picking up only soluble mHTT in the next paragraph. These are contradictory statements. S830 does, in fact, detect soluble mHTT, so should not be used as a basis for making statements about the role of aggregates in HD pathogenesis. The authors state that it is widely postulated that Htt aggregates are the source of HD toxicity. However, it has been well described that neurons with aggregates live longer than neurons without aggregates, suggesting a protective role of aggregation. Additionally, multiple mouse models with robust brain aggregation, shuch as the YAC128 C6R and short stop models, have transgene alterations that make them non-pathogenic i.e. no neurodegeneration, not HD-like behavior phenotype, but still result in robust brain HTT aggregates. So, it's been well established in the field for a long time that aggregation is not the driver of disease in HD.

We thank the reviewer for pointing out this oversight. We now revise to

“S830 …which specifically detects soluble and aggregated mutant Htt”.

This is consistent with our experiments that S830 detects soluble mHtt at 2 months, and aggregated mHtt starting at 4 months.

With regard to the mHtt aggregation being pathogenic, innocuous or beneficial, our manuscript does not have experiments for investigating this question. The only relevant sentence in this manuscript “…It is widely postulated that mHtt aggregates are the source of HD toxicity…” is now eliminated. We claim no further than the fact that BAC226Q mice exhibit age-dependent and progressive development of cytosolic and neuritic mHtt aggregations as well as nuclear inclusions, which are hallmarks of human HD patients.

– The authors state that because they observe robust disease in their model and the full complement of mouse Htt is still present, this demonstrates that HD is a GOF disease, not LOF. First, this is not a novel finding. Like several of the things in the Discussion section, it is common to other FL mouse models of HD, thus, not novel. This includes things like onset of cognitive and psychiatric behavioral abnormalities prior to motor abnormalities (Seen in YAC128, BACHD, Humanized mice, Q175), or robust disease without overt dependence on cleavage (hard to tell without fragment analysis, but presumably not different from BACHD). Furthermore, mHTT is known to sequester wtHTT or do its job poorly and potentially competitively, so just because the wtHTT is still there, doesn't' meant that the mHTT can't have some dominant negative effect and result in LOF toxicities. Thus, this study clarifies GOF vs LOF.

Indeed, the question of gain-of-function vs. loss-of-function, dominant negative effect of mHtt is not new and has been explored by colleagues in the field. However, it is still important and especially relevant to gene editing as a therapeutic method. This is the reason we discuss it here. After considering several lines of evidence, we therefore revise the manuscript as follows:

“Another important question is whether mHtt has a dominant gain-of-toxic function, or a combination of loss of function of the wild type allele. In human patients, rare cases of disrupting one Htt allele didn’t develop abnormality (Ambrose et al., 1994), patients with homozygous mutant Htt alleles develop HD similar to heterozygous carriers (Dürr et al., 1999; Kremer et al., 1994; Squitieri et al., 2003; Wexler et al., 1987). In BAC226Q, mHTT was not overexpressed, and the two alleles of the wild type HTT exist. The fact that BAC226Q mice developed such robust HD-like phenotype, and that wild type Htt knockout in adult mice didn’t induce HD-like phenotypes (Leavitt et al., 2020; Wang et al., 2016), seem to give more support to the “gain-of-function” hypothesis. An interesting future experiment will be to put BAC226Q in Htt^+/-^ and Htt^-/-^ background, and examine whether the current HD-like phenotypes in BAC226Q are enhanced. The implication of the above discussion is whether deleting mutant HTT allele will be sufficient to benefit patients in gene-targeting as a therapeutic method.”

4. Often the figure call outs are out of order in the text. For instance, in figure 2, survival data is presented first followed by body weight data, whereas in the text, body weight data is presented prior to survival data. Please re-order either the text or the figure. This also occurs with panel 5A coming after 5B, and 6E coming after 6G in the text.

We have revised to keep consistent orders between figures and the text.

5. The introduction states that the causative gene for HD was identified 2 decades ago. This was actually 28 years; closer to 3 decades.

We have revised to spell out 28 years.

6. In the description of existing mouse model of HD, no distinction is made between KI models where only additional CAG is knocked in to the mouse Htt gene and models where the entirety of human HTT exon 1 is knocked in to the mouse gene. This is an important distinction for assessment of therapies that target exon 1 outside of the CAG tract.

We have revised to distinguish the two types of KI:

“knock-in models with expanded CAG repeats inserted into a mouse Htt exon 1 such as HdhQ72 and HdhQ150 or into a humanized exon 1 sequence such as HdhQ111, HdhQ140, N160Q, zQ175 and Q175FDN (Heng et al., 2007; Hickey et al., 2008; Levine et al., 1999; Lin et al., 2001; Liu et al., 2016; Menalled et al., 2012; Menalled et al., 2003; Shelbourne et al., 1999; Southwell et al., 2016; Wheeler et al., 1999)”

7. Please include N in figure legends 3 and 4 as are done in 2 and 5.

We have revised to include the number of animals.

8. One ref in the introduction is denoted as 4 author last names et al., while all others are only 1 name et al., or 2 names with and please change for consistency.

We have revised to the Endnote filter for Cell.

9. Weight loss Results section says that the tg mice gained weight the same as their littermates for the first 8 weeks, but the graph in 2B shows total overlap for 13 weeks (90 days).

We have revised to:

“In the developmental stage, male and female BAC226Q mice gained weights at the same rate as their non-transgenic littermates (male BAC226Q n=8; male Non Tg n=10; female BAC226Q n=8; female Non Tg n=7). Fit a growth curve model to the mice, BAC226Q and non-transgenic littermates gained weights at the same rate (male 0.381±0.060 g/day, *p*=0.7123; female 0.156±0.035 g/day, *p*=0.09).”

10. DARPP-32 staining is not a way to quantify MSNs. Please rephrase this in results. It is a marker of healthy MSNs. Expressions can be reduced and lost prior to death of cells. Even if this wasn't the case, using IOD of staining throughout the striatum doesn't quantify cells.

We have revised to “… evaluate MSNs health in BAC226Q striatum…”.